



**Emissions of intermediate-volatility and semi-volatile organic compounds from**
**domestic fuels used in Delhi, India**
Gareth J. Stewart[1], Beth S. Nelson[1], W. Joe F. Acton[2,a], Adam R. Vaughan[1], Naomi J. Farren[1],
James R. Hopkins[1,3], Martyn W. Ward[1], Stefan J. Swift[1], Rahul Arya[4], Arnab Mondal[4], Ritu
Jangirh[4], Sakshi Ahlawat[4], Lokesh Yadav[4], Sudhir K. Sharma[4], Siti S. M. Yunus[5], C. Nicholas
Hewitt[2], Eiko Nemitz[6], Neil Mullinger[6], Ranu Gadi[7], Lokesh. K. Sahu[8], Nidhi Tripathi[8],
Andrew R. Rickard[1,3], James D Lee[1,3], Tuhin K. Mandal[4] and Jacqueline F. Hamilton[1].
[1] Wolfson Atmospheric Chemistry Laboratories, Department of Chemistry, University of York, York, YO10 5DD, UK
[2] Lancaster Environment Centre, Lancaster University, Lancaster LA1 4YQ, UK
[3] National Centre for Atmospheric Science, University of York, York, YO10 5DD, UK
[4] CSIR-National Physical Laboratory, Dr. K.S. Krishnan Marg, New Delhi, Delhi 110012, India
[5] School of Water, Environment and Energy, Cranfield University, Cranfield, MK43 0AL, UK
[6] UK Centre for Ecology and Hydrology, Bush Estate, Penicuik, EH26 0QB, UK
[7] Indira Gandhi Delhi Technical University for Women, Kashmiri Gate, New Delhi, Delhi 110006, India
[8] Physical Research Laboratory (PRL), Ahmedabad 380009, India
[a] Now at: School of Geography, Earth and Environmental Sciences, University of Birmingham, B15 2TT, Birmingham, UK
**Abstract**
Biomass burning emits significant quantities of intermediate-volatility and semi-volatile
volatile organic compounds (I/SVOCs) in a complex mixture, probably containing many
thousands of chemical species. These components are significantly more toxic and have poorly
understood chemistry compared to volatile organic compounds routinely analysed in ambient
air, however quantification of I/SVOCs presents a difficult analytical challenge.
The gases and particles emitted during the test combustion of a range of domestic solid fuels
collected from across New Delhi were sampled and analysed. Organic aerosol was collected
onto Teflon (PTFE) filters and residual low-volatility gases were adsorbed to the surface of
solid-phase extraction (SPE) disks. A new method relying on accelerated solvent extraction
(ASE) coupled to comprehensive two-dimensional gas chromatography with time-of-flight
mass spectrometry (GC×GC-ToF-MS) was developed. This highly sensitive and powerful
analytical technique enabled over 3000 peaks from I/SVOC species with unique mass spectra
to be detected. 15-100 % of gas-phase emissions and 7-100 % of particle-phase emissions were
characterised. The method was analysed for suitability to make quantitative measurements of
I/SVOCs using SPE disks. Analysis of SPE disks indicated phenolic and furanic compounds
were important to gas-phase I/SVOC emissions and levoglucosan to the aerosol phase. Gas-





and particle-phase emission factors for 21 polycyclic aromatic hydrocarbons (PAHs) were
derived, including 16 compounds listed by the US EPA as priority pollutants. Gas-phase
emissions were dominated by smaller PAHs. New emission factors were measured (mg kg$^{-1}$)
for PAHs from combustion of cow dung cake (615), municipal solid waste (1022), crop residue
(747), sawdust (1236), fuel wood (247), charcoal (151) and liquified petroleum gas (56).
The results of this study indicate that cow dung cake and municipal solid waste burning are
likely to be significant PAH sources and further study is required to quantify their impact,
alongside emissions from fuel wood burning.
**Introduction**
Biomass burning is one of the most important global sources of trace gases and particles to the
atmosphere (Simoneit, 2002; Chen et al., 2017; Andreae, 2019). Emissions of volatile organic
compounds (VOCs) and particulate matter (PM) are of interest due to their detrimental impact
on air quality. VOCs react to form ozone and secondary organic aerosol (SOA), and contribute
0-95 Tg yr$^{-1}$ of SOA yearly (Shrivastava et al., 2017). Estimates of VOCs from burning often
do not include many intermediate-volatility and semi-volatile organic compounds (I/SVOCs).
Wildfires emit significant quantities of organic matter over regions such as the USA, the
Mediterranean, South East Asia and Australia (Liu et al., 2017; Barboni et al., 2010; Kiely et
al., 2019; Guérette et al., 2018) and residential combustion leads to substantial organic
emissions in the developing world (Streets et al., 2003).
I/SVOCs are an important class of air pollutant due to their contribution to aerosol formation
(Bruns et al., 2016; Lu et al., 2018). IVOCs have an effective saturation concentration of 300-
3,000,000 μg m$^{-3}$ and are predominantly in the vapour phase. Once oxidised their lower
volatility products can partition into the aerosol phase (Donahue et al., 2006). SVOCs have
effective saturation concentrations of 0.3-300 μg m$^{-3}$ (Donahue et al., 2012) and can partition
between the gas and particle phases. Many studies have focused on I/SVOCs emitted from a
range of sources due to their impact on aerosol formation (Robinson et al., 2007; Zhao et al.,
2014; Zhao et al., 2015; Cross et al., 2015; Pereira et al., 2018). I/SVOCs have been shown to
contribute significantly to emissions from biomass burning (Stockwell et al., 2015; Koss et al.,
2018). Global I/SVOC emissions to the atmosphere from biomass burning were estimated to
be ~ 54 Tg yr$^{-1}$ from 2005-2008 (Hodzic et al., 2016) with I/SVOCs contributing in the range
8-15.5 Tg yr$^{-1}$ to SOA (Cubison et al., 2011; Hodzic et al., 2016). SOA formation from
combustion of beech fuel wood was shown to be dominated by 22 compounds, with phenol,



naphthalene and benzene contributing up to 80 % of the observed SOA (Bruns et al., 2016).
However, the effect of atmospheric of aging on I/SVOCs still remains poorly understood (Liu
et al., 2017; Decker et al., 2019; Sengupta et al., 2020).
Residential combustion, agricultural crop residue burning, and open municipal solid waste
burning in the developing world are large, poorly characterised pollution sources with the
potential to have a significant impact on local and regional air quality, impacting human health
(Venkataraman et al., 2005; Jain et al., 2014; Wiedinmyer et al., 2014). Hazardous indoor air
pollution from combustion of solid fuels has been shown to be the most important factor from
a range of 67 environmental and lifestyle risk factors causing disease in South Asia (Lim et al.,
2012). Recent studies focussed on source apportionment of ambient VOC concentrations in
Delhi have shown ground-level concentrations to be predominantly traffic related, with smaller
contributions from solid fuel combustion (Stewart et al., 2020; Wang et al., 2020). Despite this,
nearly 76 % of rural Indian households are dependent on solid biomass for their cooking needs
(Gordon et al., 2018) with biofuels such as fuel wood, cow dung cake and crop residue being
used. Combustion often takes place indoors without efficient emission controls which
significantly increases the mean household concentration of pollutants, particularly particulate
matter with a diameter less than 2.5 $\mu$m ($PM_{2.5}$). Studies have shown mean 24 h concentrations
of $PM_{2.5}$ in kitchens to be in excess of 500 $\mu$g m$^{-3}$ (Balakrishnan et al., 2013), with 4-8 times
ambient concentration enhancement of polycyclic aromatic hydrocarbons (PAHs) close to the
stove during cooking (Bhargava et al., 2004). This is significantly larger than the 40 $\mu$g m$^{-3}$
Indian National Air Quality Standard. For comparison, the mean population weighted $PM_{2.5}$
level in Delhi, Chennai, Hyderabad and Mumbai from 2015-2018 was 72 $\mu$g m$^{-3}$ and the global
mean 20 $\mu$g m$^{-3}$ (Chen et al., 2020), with various sources also leading to elevated levels of
PAHs in cities like Delhi (Elzein et al., 2020). The health effects from this are significant, with
premature deaths in India from exposure to ambient and household air pollution estimated to
be over 2 million (Lallukka et al., 2017).
Few detailed studies have been conducted examining the composition of I/SVOC emissions
from sources relevant to South Asia. One study examined gas- and particle-phase emissions
from coconut leaves, rice straw, cow dung cake, biomass briquettes and jackfruit branches in
Bangladesh with samples analysed by ion chromatography (IC), organic/elemental carbon
(OC/EC) and gas-chromatography coupled to mass spectrometry (GC-MS) to produce
emission factors and examine molecular markers (Sheesley et al., 2003). Another study
examined emissions of $PM_{2.5}$, OC/EC, metals and organics from motorcycles, diesel- and



gasoline-generators, agricultural pumps, municipal solid waste burning, cooking fires using
fuel wood and cow dung cake, crop residue burning and brick kilns in Nepal (Jayarathne et al.,
2018). Lack of knowledge regarding major pollution sources hinders our ability to predict air
quality, but also the development of effective mitigation strategies for air pollution which leads
to health impacts ranging from respiratory illness to premature death (Brunekreef and Holgate,
2002). This results in many people living with high levels of air pollution (Lelieveld et al.,
2015; Cohen et al., 2005) and 13 Indian cities ranking amongst the top 20 cities in the world
with the highest levels of ambient $PM_{2.5}$ pollution, based on available data (Gordon et al.,

107   2018).

Early biomass burning studies used filters to target aerosol and sorbent tubes or polyurethane
styrene-divinylbenzene (PUF/XAD/PUF) cartridges to sample gaseous species followed by
solvent extraction and analysis by GC-MS (McDonald et al., 2000; Schauer et al., 2001; Hays
et al., 2002; Mazzoleni et al., 2007; Dhammapala et al., 2007; Singh et al., 2013; Jordan and
Seen, 2005; Pettersson et al., 2011; Sheesley et al., 2003). Detailed studies have focussed on
quantifying the composition of the particulate matter from burning by extracting aerosol
samples, followed by analysis by GC-MS (Oros and Simoneit, 2001b, a; Fine et al., 2001; Oros
et al., 2006; Jayarathne et al., 2018). Many studies have been carried out to measure emission
factors of PAHs from burning, such as detailed measurements of up to 133 PAHs (Samburova
et al., 2016) and time-resolved PAH measurements (Eriksson et al., 2014). PAH emission
factors have been measured for coal (Chen et al., 2005; Lee et al., 2005; Geng et al., 2014) oil
(Rogge et al., 1997), fuel woods (McDonald et al., 2000; Simoneit, 2002; Hosseini et al., 2013;
Jimenez et al., 2017; Geng et al., 2014), peat (Iinuma et al., 2007), tyres (Iinuma et al., 2007),
domestic waste (Sidhu et al., 2005; Kakareka et al., 2005), cow dung cake (Gadi et al., 2012;
Singh et al., 2013; Tiwari et al., 2013), sawdust briquette (Kim Oanh et al., 2002) and crop
residue (Jenkins et al., 1996; Lu et al., 2009; Gadi et al., 2012; Singh et al., 2013; Wei et al.,
2014; Kim Oanh et al., 2015; Wiriya et al., 2016). Measurements of I/SVOCs in both gas- and
particle-phase samples using conventional GC-MS presents a difficult analytical challenge, due
to the exponential growth of potential isomers with carbon number which can result in a large
number of coeluting peaks (Goldstein and Galbally, 2007).
The high resolution of two-dimensional gas chromatography (GCxGC) has been demonstrated
as an ideal technique to overcome the issue of peak coelution in one-dimensional gas
chromatography and has been used to analyse complex ambient samples in the gas (Lewis et
al., 2000; Xu et al., 2003; Dunmore et al., 2015; Lyu et al., 2019a) and particle phases



(Hamilton et al., 2004; Lyu et al., 2019b; Lyu et al., 2019c). GCxGC has recently shown
hundreds of gaseous I/SVOCs released from biomass burning using adsorption-thermal
desorption cartridges or solid phase extraction (SPE) disks (Hatch et al., 2015; Hatch et al.,
2018). The particle phase has also been targeted by extracting samples from PTFE or quartz
filters (Hatch et al., 2018; Jen et al., 2019), with the latter study quantifying 149 organic
compounds which accounted for 4-37 % of the total mass of organic carbon. The process used
by Hatch et al. (2018) demonstrated high recoveries of non-polar species from PTFE filters,
with lower recoveries from SPE disks. This study highlighted the need for further evaluation
of samples collected onto PTFE filters and SPE disks, ideally improving the method to remove
undesirable steps such as trimethylsilyation derivatisation, the use of pyridine and centrifuging
which led to high evaporative losses. The need to develop improved sampling and
measurement techniques for I/SVOCs has been highlighted as these species often do not
transmit quantitatively through the inlet and tubing when measured using online gas-phase
techniques (Pagonis et al., 2017).
In this study we develop a more efficient extraction step for the SPE/PTFE technique, allowing
high recoveries of non-polar I/SVOCs collected from burning typical domestic fuels used in
Northern India. The technique is used to identify many I/SVOCs in burning samples, examined
for quantification of I/SVOCs from burning and used to develop emission factors for selected
PAHs.

**Sample collection and burning facility**

The state of New Delhi was gridded (0.05º×0.05º) and samples collected from across the state
(see Figure 1). Samples were stored in a manner akin to local practices prior to combustion, to
ensure that the moisture content of fuels were similar to those burnt across the state. A range
of solid biomass fuels were collected which included 17 fuel wood species, cow dung cake,
charcoal and sawdust (see Table 1). Three crop residue samples were collected and consisted
of dried stems from vegetable plants such as cabbage (*Brassica spp*) and aubergines (*Solanum*
*melongena*) as well as coconut husk (*Cocos nucifera*). Municipal solid waste samples were
collected from Bhalaswa, Ghazipur and Okhla landfill sites. A low-cost LPG stove was also
purchased to allow direct comparison to other combustion sources.
Samples were burnt at the CSIR-National Physical Laboratory (NPL) New Delhi under
controlled conditions using a combustion dilution chamber that has been well described
previously (Venkataraman et al., 2002; Saud et al., 2011; Saud et al., 2012; Singh et al., 2013).





In summary, 200 g of dry fuel was rapidly heated to spontaneous ignition with emissions driven
into a hood and up a flue by convection to allow enough dilution, cooling and residence time
to achieve the quenching of typical indoor environments. This process was designed to
replicate the immediate condensational processes that occur in smoke particles approximately
5-20 mins after emission, yet prior to photochemistry which may change composition (Akagi
et al., 2011). A low volume sampler (Vayubodhan Pvt.Ltd) was used to collect particulates and
low volatility gases passing from the top of the flue through a chamber with a flow rate of 46.7
L min$^{-1}$. As detailed in Table 1, 30 samples from a range of fuel types were burnt, and 8 blank
samples were collected (see the Supplementary Information S1 for an example burn and filter
samples collected from different sources). Prior to sample collection, SPE disks (Resprep, $C_{18}$,
47 mm) were prewashed with 2 x 5 mL acetone (Fisher Scientific analytical reagent grade),
and 1 x 5 mL methanol (Sigma-Aldrich HPLC grade), then packed in foil and sealed in airtight
bags. Samples were collected onto a PTFE filter (Cole-Parmer, 47 mm, 1.2 μm pore size)
placed on top of an SPE disk in a filter holder (Cole-Parmer, 47 mm, PFA) for 30 mins at a
flow rate of 6 L min$^{-1}$, maintained by a mass flow controller (Alicat 0-20 SLM) connected to a
pump. Samples were removed from the filter holder immediately after the experiment and
wrapped in foil, placed inside an airtight bag and stored at – 20 °C. Samples were then
transported to the UK for analysis using an insulated container containing dry ice via. air freight
and stored at – 20 °C for around 2 months prior to analysis.

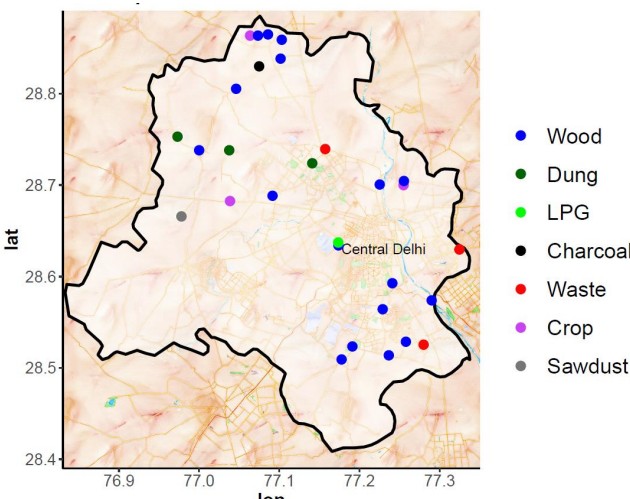


Figure 1. Locations across New Delhi used for the local surveys into fuel use and sample
collection. Map produced using tiles from Stamen maps.





Table 1. Types of sample collected where $n$ = number of samples burned, SPE and PTFE = number of
blank corrected peaks detected on SPE disks and PTFE filters, respectively.

| Fuel woods | $n$ | SPE | PTFE | Other | $n$ | SPE | PTFE |
|---|---|---|---|---|---|---|---|
| Plywood | 1 | 149 | 530 | Cow dung cake | 3 | 1295 | 1604 |
| *Azadirachta indica* | 1 | 562 | 880 | *Cocos nucifera* | 1 | 614 | 1182 |
| *Morus spp* | 1 | 811 | 1108 | Charcoal | 1 | 453 | 211 |
| *Shorea spp* | 1 | 283 | 326 | Sawdust | 1 | 1113 | 1417 |
| *Ficus religiosa* | 1 | 504 | 652 | Waste | 3 | 980 | 1181 |
| *Syzygium spp* | 1 | 680 | 529 | LPG | 1 | - | 0 |
| *Ficus spp* | 1 | 277 | 247 | Blank | 8 | - | - |
| *Vachellia spp* | 1 | 702 | 753 | Cow dung cake mix | 1 | 932 | 1200 |
| *Dalbergia sissoo* | 1 | 483 | 561 | *Brassica spp* | 1 | 656 | 463 |
| *Ricinus spp* | 1 | 424 | 125 | *Solanum melongena* | 1 | 280 | 551 |
| *Holoptelea spp* | 1 | 276 | 263 | | | | |
| *Saraca indica* | 1 | 517 | 445 | | | | |
| *Pithecellobium spp* | 1 | 527 | 159 | | | | |
| *Eucalyptus spp* | 1 | 211 | 77 | | | | |
| *Melia azedarach* | 1 | 434 | 166 | | | | |
| *Prosopis spp* | 1 | 237 | 113 | | | | |
| *Mangifera indica* | 1 | 360 | 546 | | | | |


**Sample extraction**

SPE disks and PTFE filters were spiked with an internal standard (50 μL at 20 μg mL$^{-1}$)
containing 6 deuterated PAHs (1,4-Dichlorobenzene-d$_4$, naphthalene-d$_8$, acenaphthene-d$_{10}$,
phenanthrene-d$_{10}$, chrysene-d$_{12}$, perylene-d$_{12}$; EPA 8270 Semivolatile Internal Standard Mix,
2000 μg mL$^{-1}$ in DCM) to result in a final internal standard concentration of 1 μg mL$^{-1}$ in
solution. The solvent from the internal standard was allowed to evaporate and then SPE disks
and PTFE filters were cut and extracted into EtOAc using accelerated solvent extraction (ASE
350, Dionex, ThermoFisher Scientific). Extractions were performed at 80 °C and 1500 psi for
three 5 min cycles. After each cycle, the cell was purged for 60 seconds into a sample collection
vial. Samples were then reduced from 15 mL to 0.90 mL over a low flow of N$_2$ in an ice bath
over a period of 6-8 hours (Farren et al., 2015). Samples were then pipetted (glass Pasteur) to
sample vials (Sigma-Aldrich, amber glass, 1.5 mL), with ASE vials rinsed with 2 × 50 μL
washes of EtOAC, then added to the sample vial and sealed (Agilent 12 mm cap,
PTFE/silicone/PTFE). The mass of the sample vial and cap for each sample was measured
before and after to determine the exact volume of solvent in each sample. Extracts were frozen
prior to analysis to reduce evaporative losses.

**Methods for organic composition analysis**

GCxGC-ToF-MS: PTFE samples were analysed using GCxGC-ToF-MS (Leco Pegasus BT
4D) using a splitless injection (1 μL injection, 4mm taper focus liner, SHG 560302). The



primary dimension column was a RXI-5SilMS (Restek, 30 m × 0.25 μm × 0.25 mm) connected
to a second column of RXI-17SilMS (Restek, 0.25 μm × 0.25 mm, 0.17 m primary GC oven,
0.1 m modulator, 1.42 m secondary oven, 0.31 m transfer line) with a He flow of 1.4 mL min$^{-1}$
. The primary oven was held at 40 $^o$C for 1 min then ramped at 3 $^o$C min$^{-1}$ to 322 $^o$C where it
was held for 3 min. The secondary oven was held at 62 $^o$C for 1 min then ramped at 3.2 $^o$C to
190 $^o$C after which it was ramped at 3.6 $^o$C min$^{-1}$ to 325 $^o$C and held for 19.5 mins. The inlet
was held at 280 $^o$C and the transfer line at 340 $^o$C. A 5 s cryogenic modulation was used with
a 1.5 s hot pulse and 1 s cool time between stages. Using two separate wash vials, the syringe
(10 μL Gerstel) was cleaned prior to injection with two cycles of 3 × 5 μL washes in EtOAc
and rinsed post injection with two cycles of 2 × 5 μL washes in EtOAc. Samples with high
concentrations of levoglucosan were reanalysed using a faster method, injected split (75:1 and
125:1) with the primary oven held at 40 $^o$C for 1 min, then ramped at 10 $^o$C min$^{-1}$ to 220 $^o$C.
The secondary oven was held at 62 $^o$C for 1 min and then ramped at 10 $^o$C min$^{-1}$ to 245 $^o$C.
SPE samples were injected split (10:1) and analysed with a shorter analysis time with the
primary oven held at 40 $^o$C for 1 min then ramped at 3 $^o$C min$^{-1}$ to 202 $^o$C where it was held for
4 seconds. The secondary oven was held at 62 $^o$C for 1 min then ramped at 3.2 $^o$C min$^{-1}$ to 235
$^o$C. A 75:1 split injection was used for quantitation of concentrations outside of the detector
response range for furans, phenolics, benzaldehydes, naphthalenes and benzonitrile. Peaks
were assigned through comparison of retention times with known standards and comparison
with the National Institute of Standards and Technology (NIST) mass spectral library. Peaks
with no genuine standard available were tentatively identified if the NIST library hit was >
800. The uncertainty in this approach has been shown to be low, with the probability of
incorrect identification being around 30 % for hits between 800-900 and 14 % for matches
above 900 (Worton et al., 2017). Integration was carried out within the ChromaTOF 5.0
software package (Leko, 2019). Calibration was performed using a 6-point calibration using
either a linear or second-order polynomial fit covering the ranges 0.1-2.5 μg ml$^{-1}$ (splitless),
0.5-15 μg ml$^{-1}$ (10:1 split), 15-400 μg ml$^{-1}$ (75:1 split) and 400-800 μg ml$^{-1}$ (125:1 split). Eight
blank measurements were made at the beginning and end of the day by passing air from the
chamber (6 L min$^{-1}$ for 30 mins) through the filter holder containing PTFE filters and SPE disks
(see the Supplementary Information S2 for examples of blank chromatograms). Blank
corrections were applied by calculating the average blank value for each compound using blank
samples collected using the same sample collection parameters as real samples before and after
the relevant burning experiments.



PTR-ToF-MS:   Online   measurements   of   naphthalene,   methylnaphthalenes   and
dimethylnaphthalenes  were  made  using  a  proton  transfer  reaction-time  of  flight-mass
spectrometer PTR-ToF-MS (PTR 8000; Ionicon Analytik, Innsbruck) and assigned as masses
129.058,  143.08  and  157.097,  respectively.  Additional  details  of  the  PTR-ToF-MS  from
Physical  Research  Laboratory  (PRL),  Ahmedabad  used  in  this  study  are  given  in  previous
papers (Sahu and Saxena, 2015; Sahu et al., 2016). A ¼ inch OD PFA sample line ran from the
top  of  the  flue  to  the  instrument  which  was  housed  in  an  air-conditioned  laboratory  with  a
sample flow rate of 4.3 L min$^{-1}$. The sample air was diluted either 5 or 6.25 times into zero air,
generated by passing ambient air (1 L min$^{-1}$) through a heated platinum filament at 550 °C,
before entering the instrument with an inlet flow of 250 ml min$^{-1}$. The instrument was operated
with a reduced electric field strength ($E/N$, where $N$ is the buffer gas density and $E$ is the electric
field strength) of 120 Td. The drift tube temperature was 60 °C with a pressure of 2.3 mbar and
560 V applied across it.
Calibrations  of  the  PTR-ToF-MS  were  performed  twice  a  week  using  a  gas  calibration  unit
(Ionicon  Analytik,  Innsbruck).  The  calibration  gas  (Apel-Riemer  Enironmental  Inc.,  Miami)
contained  18  compounds:  methanol,  acetonitrile,  acetaldehyde,  acetone,  dimethyl  sulphide,
isoprene,  methacrolein,  methyl  vinyl  ketone,  2-butanol,  benzene,  toluene,  2-hexanone,  $m$-
xylene, heptanal, α-pinene, 3-octanone and 3-octanol at 1000 ppb (±5 %) and β-caryophyllene
at 500 ppb (±5 %). This standard was dynamically diluted into zero air to provide a 6-point
calibration. The normalised sensitivity (ncps/ppbv) was then determined for all masses using a
transmission curve derived from these standard compounds (Taipale et al., 2008).
Mass calibration and peak fitting of the PTR-ToF-MS data were performed using PTRwid
software (Holzinger, 2015). Count rates (cps) of each mass spectral peak were normalised to
the primary ion ($H_3O^+$) and water cluster ($H_3O.H_2O$)$^+$ peaks and mixing ratios were then
determined for each mass using the normalised sensitivity (ncps). Where compounds known to
fragment in the PTR-ToF-MS were identified, the mixing ratio of these species was calculated
by summing parent ion and fragment ion mixing ratios. Before each burning study ambient air
was sampled to provide a background for the measurement.
**Quantification of recovery and breakthrough**
Standards were used for 136 species (see Figure 2) including two commercially available
standard mixes containing 33 alkanes (C$_7$-C$_{40}$ saturated alkane standard, certified 1000 µg m$^{-1}$
in hexane, Sigma Aldrich 49452-U) and 64 semi volatiles (EPA CLP Semivolatile Calibration





Mix, 1000 μg mL$^{-1}$ in DCM:benzene 3:1, Sigma Aldrich 506508). Further standards were
produced in-house, by dissolving high quality standards (> 99 % purity), for a range of
additional species also found in samples including nitrogen containing VOCs (NVOCs),
furans, alkyl-substituted monoaromatics, oxygenated aromatics, ketones, aldehydes, methoxy
phenols, aromatic acids, PAHs and levoglucosan (see Table 2). Stock solutions of around 1000
μg mL$^{-1}$ were prepared by dissolving 0.01 g into 10 mL EtOAc. Polar components, such as
levoglucosan, were dissolved into MeOH for stock solutions and those not soluble at room
temperature were heated and pipetted using hot pipette tips to make quantitative dilutions.
Six separate PTFE filters and SPE disks were spiked with the standard solution containing 136
compounds (50 μL at 20 μg mL$^{-1}$), extracted and analysed. Recovery levels were calculated by
comparing the signal to direct injection of the diluted standards to the GCxGC-ToF-MS. The
recoveries are shown in Table 2. SPE disks showed poor recoveries ($S_{rec}$) of *n*-nonane to *n*-
tridecane and C$_2$ substituted monoaromatics, likely due to volatilisation of these more-volatile
components. Poorer recoveries were also observed of nitroanilines and levoglucosan. Non-
polar species showed good recoveries, with high recoveries of C$_{14}$-C$_{20}$ alkanes, furans, phenols,
chlorobenzenes and PAHs. PTFE filters demonstrated high recoveries ($P_{rec}$) of PAHs with more
than three rings in their structure (81.6-100 %). Recoveries were low, or zero, for volatile
components with boiling points < 200 °C, indicating no retention, which is consistent with the
method being well-suited to target the aerosol phase. The recoveries of non-polar species in
EtOAc from SPE disks were higher than those reported into MeOH (Hatch et al., 2018).

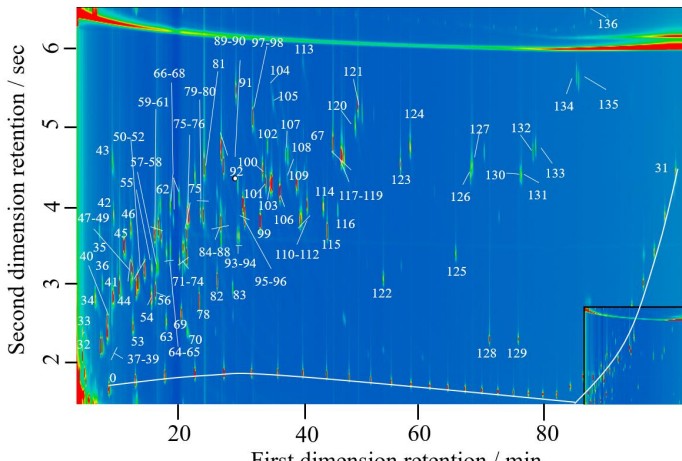

Figure 2. GCxGC-ToF-MS chromatogram of a mixed standard, numbered according to species listed
in Table 2.



Table 2. Species used in calibration where Comp No. refers to the peak number in Figure 2, $Q_{ms}$ = split method used for SPE quantitation, $Q_{mp}$ = split method used for PTFE quantitation, S = splitless method, $S_{rec}$ = % recovery SPE, $P_{rec}$ = % recovery PTFE, [a] = Sigma-Aldrich $n$-alkanes standard, [b] = Sigma-Aldrich semivolatiles standard, [c] = Sigma-Aldrich deuterated internal standard, [d] = in-house solution and **-** = not measured either due to being outside of SPE method range or due to volatilisation from PTFE filters. Slight over-recoveries of > 100 % are reported as 100 % and accounted for in blank subtractions.

| Comp No. | Species | $Q_{ms}$ | $Q_{mp}$ | $S_{rec}$ | $P_{rec}$ | Comp No. | Species | $Q_{ms}$ | $Q_{mp}$ | $S_{rec}$ | $P_{rec}$ |
|---|---|---|---|---|---|---|---|---|---|---|---|
| **Alkane** | | | | | | **NVOC** | | | | | |
| 0 | $n$-Nonane [a] | 10:1 | S | 60.0 | - | 32 | Pyridine [d] | 10:1 | S | 75.1 | - |
| 1 | $n$-Decane [a] | 10:1 | S | 77.6 | 19.5 | 33 | $n$-Nitrosodimethylamine [b] | 10:1 | S | - | - |
| 2 | $n$-Undecane [a] | 10:1 | S | 100 | 57.2 | 44 | 2,3-lutidine [d] | 10:1 | S | 99.4 | - |
| 3 | $n$-Dodecane [a] | 10:1 | S | 85.7 | 22.0 | 46 | Benzonitrile [d] | 75:1 | S | 86.9 | - |
| 4 | $n$-Tridecane [a] | 10:1 | S | 91.4 | 75.0 | 57 | $n$-Nitrosodipropylamine [b] | 10:1 | S | 100 | - |
| 5 | $n$-Tetradecane [a] | 10:1 | S | 97.8 | 97.8 | 62 | Nitrobenzene [b] | 10:1 | S | 88.5 | - |
| 6 | $n$-Pentadecane [a] | 10:1 | S | 99.7 | 92.3 | 67 | 2-Nitrophenol [b] | 10:1 | S | 100 | - |
| 7 | $n$-Hexadecane [a] | 10:1 | S | 100 | 100 | 68 | Pyrrole 2-carbonitrile [d] | 10:1 | S | - | - |
| 8 | $n$-Heptadecane [a] | 10:1 | S | 100 | 98.0 | 77 | 4-chloroanaline [b] | 10:1 | S | 7.78 | - |
| 9 | $n$-Octadecane [a] | 10:1 | S | 100 | 99.9 | 98 | 2-Nitroanaline [b] | 10:1 | S | 100 | - |
| 10 | $n$-Nonadecane [a] | 10:1 | S | 100 | 98.9 | 102 | 2,6-dinitrotoluene [b] | 10:1 | S | 99.9 | - |
| 11 | $n$-Eicosane [a] | 10:1 | S | 100 | 96.8 | 105 | 3-Nitroanaline [b] | 10:1 | S | 34.2 | - |
| 12 | $n$-Heneicosane [a] | 10:1 | S | - | 100 | 107 | 2,4-Dinitrotoluene [b] | 10:1 | S | 100 | - |
| 13-23 | $n$-Docosane [a] – $n$-Dotriacontane [a] | 10:1 | S | - | 100 | 108 | 4-Nitrophenol [b] | 10:1 | S | - | - |
| 24 | $n$-Tritriacontane | - | - | - | 96.5 | 112 | Azobenzene [b] | 10:1 | | 100 | 100 |
| 25 | $n$-Tetratriacontane | - | - | - | 78.9 | 113 | $p$- Nitroaniline [b] | 10:1 | S | 64.5 | - |
| 26 | $n$-Pentatriacontane | - | - | - | 58.3 | 121 | Caffeine [d] | 10:1 | S | - | - |
| 27 | $n$-Hexatriacontane | - | - | - | 49.9 | **Aromatics** | | | | | |
| 28 | $n$-Heptatriacontane | - | - | - | 35.4 | 37 | Ethylbenzene [d] | 10:1 | S | 44.6 | - |
| 29 | $n$-Octatriacontane | - | - | - | 32.1 | 38 | $m$-Xylene [d] | 10:1 | S | 34.5 | - |
| 30 | $n$-Nonatriacontane | - | - | - | 29.1 | 39 | $o$-Xylene [d] | 10:1 | S | 32.4 | - |
| 31 | $n$-Tetracontane | - | - | - | 27.9 | 40 | Styrene [d] | 10:1 | S | 58.4 | - |
| **PAH** | | | | | | 69 | Pentylbenzene [d] | 10:1 | S | 99.0 | 24.4 |
| 76 | Naphthalene [b,c] | 75:1 | S | 93.9 | 37.1 | 82 | Pentamethylbenzene [d] | 10:1 | S | 68.6 | 39.5 |
| 81 | Quinoline [d] | 10:1 | S | 28.6 | - | **Halogenated** | | | | | |
| 87 | 2-Methylnapthalene [b] | 75:1 | S | 90.8 | 72.4 | 48 | 2-Chlorophenol [b] | 10:1 | S | 100 | - |
| 89 | Indole [d] | 10:1 | S | 81.6 | - | 50 | 1,3-Dichlorobenzene [b] | 10:1 | S | 85.5 | - |
| 90 | Azulene [d] | 10:1 | S | 38.5 | - | 51 | 1,4-Dichlorobenzene [b,c] | 10:1 | S | 87.2 | - |
| 91 | 1(3H)-Isobenzofuranone [d] | 10:1 | S | 100 | - | 52 | 1,2-Dichlorobenzene [b] | 10:1 | S | 70.3 | - |
| 96 | Biphenyl [d] | 10:1 | S | 99.5 | 75.0 | 56 | Hexachloroethane [b] | 10:1 | S | 83.7 | - |
| 97 | 1,4-Naphthoquinone [d] | 10:1 | S | 100 | - | 74 | 2,4-Dichlorophenol [b] | 10:1 | S | 100 | 83.9 |
| 99 | 2,3-Dimethylnaphthalene [d] | 10:1 | S | 100 | - | 75 | 1,2,4-trichlorobenzene [b] | 10:1 | S | 85.6 | - |
| 100 | Acenaphthylene [b] | 10:1 | S | 98.5 | 84.1 | 78 | Hexachlorobutadiene [b] | 10:1 | S | 61.6 | - |
| 103 | Acenapthene [b,c] | 10:1 | S | 100 | 88.2 | 83 | Hexachlorocyclopentadiene [b] | 10:1 | S | 100 | - |
| 106 | Dibenzofuran [b] | 10:1 | S | 100 | 86.4 | 88 | 4-Chloro-3-methylphenol [b] | | S | 90.8 | - |
| 109 | Fluorene [b] | 10:1 | S | 100 | 86.0 | 93 | 2,4,6-Trichlorophenol [b] | 10:1 | S | 95.8 | - |
| 117 | 9H-Fluoren-9-one [d] | 10:1 | S | 100 | 100 | 94 | 2,4,5-Trichlorophenol [b] | 10:1 | S | 100 | - |
| 118 | Phenanthrene [b] | 10:1 | S | 100 | 96.7 | 95 | 2-Chloronapthalene [b] | 10:1 | S | 99.6 | - |
| 119 | Anthracene [b] | 10:1 | S | 98.6 | 95.9 | 110 | 4-Chlorophenylphenylether [b] | 10:1 | S | 100 | - |
| 120 | Carbazole [b] | 10:1 | S | 100 | 85.2 | 114 | 4-Bromophenylphenylether [b] | 10:1 | S | 100 | - |
| 123 | Fluoranthene [b] | 10:1 | S | 100 | 97.2 | 115 | Hexachlorobenzene [b] | 10:1 | S | 100 | - |
| 124 | Pyrene [b] | 10:1 | S | - | 100 | 116 | Pentachlorophenol [b] | 10:1 | S | 100 | - |
| 126 | Benzo(a)anthracene [b] | - | S | - | 100 | **Furans** | | | | | |




| No. | Compound | Ratio | S | Val1 | Val2 |
|---|---|---|---|---|---|
| 127 | Chrysene [b/c] | - | S | - | 100 |
| 130 | Benzo(b)fluoranthene [b] | - | S | - | 100 |
| 131 | Benzo(k)fluoranthene [b] | - | S | - | 100 |
| 132 | Benzo(a)pyrene [b] | - | S | - | 89.5 |
| 133 | Perylene-D12 [c] | - | S | - | 92.4 |
| 134 | Indeno(1,2,3-CD)pyrene [b] | - | S | - | 94.0 |
| 135 | Dibenz(A,H)anthracene [b] | - | S | - | 92.9 |
| 136 | Benzo(G,H,I)perylene [b] | - | S | - | 96.6 |
| **Oxygenated aromatics** | | | | | |
| 41 | Anisole [d] | 10:1 | S | 20.4 | - |
| 42 | p-Benzoquinone [d] | 10:1 | S | 94.8 | - |
| 45 | Benzaldehyde [d] | 10:1 | S | 82.8 | - |
| 47 | Phenol [b] | 75:1 | S | 100 | - |
| 55 | o-Cresol [b] | 10:1 | S | 100 | - |
| 58 | p-Cresol [b] | 75:1 | S | 100 | - |
| 59 | 3-Methylbenzaldehyde [d] | 10:1 | S | 99.9 | - |
| 60 | 2-Methylbenzaldehyde [d] | 75:1 | S | 100 | - |
| 61 | 2-Methoxyphenol [d] | 75:1 | S | 100 | - |
| 64 | 2,6-Dimethylphenol [d] | 75:1 | S | 100 | 100 |
| 66 | 2,3-dimethyl-2,5-cyclohexadiene-1,4-dione [d] | 10:1 | S | 100 | - |
| 71 | 2,4-dimethylphenol [b] | 10:1 | S | 89.5 | - |
| 73 | Benzoic acid [d] | 10:1 | S | - | - |
| 79 | Mequinol [d] | 10:1 | S | 60.4 | - |
| 80 | m-Guaiacol [d] | 10:1 | S | 44.0 | - |
| 85 | Hydroquinone [d] | 10:1 | S | 34.8 | - |
| 86 | Resorcinol [d] | 10:1 | S | 76.0 | - |
| 92 | 2,6-Dimethoxyphenol [d] | 10:1 | S | 93.6 | - |

| No. | Compound | Ratio | S | Val1 | Val2 |
|---|---|---|---|---|---|
| 34 | Furfural [d] | 75:1 | S | 84.3 | - |
| 35 | Maleic anhydride [d] | 10:1 | S | 54.9 | - |
| 36 | α-Angelica lactone [d] | 10:1 | S | 52.1 | - |
| 43 | 2-5(H)-furanone [d] | 75:1 | S | 100 | - |
| **Phthalates** | | | | | |
| 101 | Dimethyl phthalate [b] | 10:1 | S | 100 | - |
| 111 | Diethyl phthalate [b] | 10:1 | S | 100 | - |
| 122 | Di-n-butyl-phthalate [b] | 10:1 | S | - | - |
| 125 | Benzyl butyl phthalate [b] | - | S | - | 92.0 |
| 128 | Bis(2-ethylhexyl)phthalate [b] | - | S | - | 97.4 |
| 129 | Di-n-octyl phthalate [b] | - | S | - | 90.6 |
| **Others** | | | | | |
| 49 | Bis(2-chloroethyl)ether [b] | 10:1 | S | 84.5 | - |
| 53 | 2-Octanone [d] | 10:1 | S | 97.0 | - |
| 54 | Bis(2-chloro-1-methylethyl)ether [b] | 10:1 | S | 100 | - |
| 63 | Nonanal [d] | 10:1 | S | 100 | 52.3 |
| 65 | Isophorone [b] | 10:1 | S | 96.4 | - |
| 70 | 1-nonanol [d] | 10:1 | S | 98.6 | - |
| 72 | Bis(2-chloroethoxy)methane [b] | 10:1 | S | 100 | - |
| 84 | Pinane diol [d] | 10:1 | S | - | - |
| 104 | Levoglucosan [d] | 10:1 | S | 0 | 70.0 |


To quantify the additional effect of breakthrough during sampling, tests were conducted for
SPE disks to examine the retention of components adsorbed to their surface when subject to an
air flow equivalent to the sample volume. SPE disks were spiked with the calibration mixture
containing 96 compounds of interest (50 μL at 20 μg mL$^{-1}$, $n = 4$) and subject to a purified air
flow of 6 L min$^{-1}$ for 30 mins. The samples were extracted and analysed, and the signal
compared with $4 \times 50$ μL spikes directly into 0.95 mL EtOAc. Figure 3 shows the relative
enhancement of unpurged over purged samples. For more volatile components a value greater
than zero was observed (Figure 3), which indicated breakthrough of the most volatile
components and indicated good retention of components with a boiling point of around 225 °C
(see the Supplementary Information S3 for breakthrough tests). Concentrations measured for
*n*-alkanes on SPE disks were also compared with concurrent measurements made during
burning experiments using online thermal-desorption two-dimensional gas chromatography
coupled to a flame ionisation detector. The measured concentrations for *n*-alkanes from *n*-
nonane to *n*-dodecane were compared using both techniques, with measured concentrations



similar for *n*-undecane/*n*-dodecane (bp = 216 °C, see the Supplementary Information S4) but
not the smaller alkanes. This was interpreted to indicate little breakthrough for components less
volatile than *n*-dodecane. These findings are in line with the US EPA certified methods for
Resprep SPE disks (525.1, 506, 550.1, and 549.1), when used to quantitatively analyse drinking
water, which show their suitability for quantitative measurement of species with a molecular
weight of around naphthalene/acenaphthylene (bp = 218-280 °C). These results indicate that
for more volatile species with boiling points below 250 °C, SPE disks can only be used to make
qualitative measurements at these sample times and flow rates. Such qualitative information is
highly complementary to quantitative measurements using other, less specific, techniques, such
as PTR-ToF-MS, where it can assist in identification of the contributors to *m/z* ions.

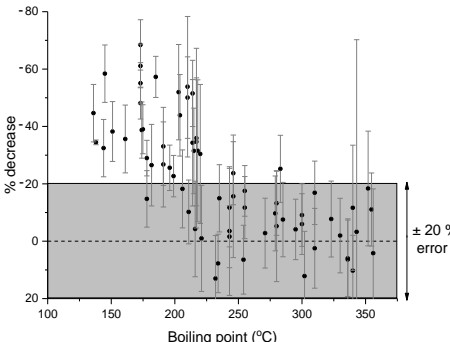


Figure 3. Relative reduction of purged over unpurged samples, presented as a percentage decrease of
purged to unpurged signal. The standard deviation of replicate measurements is indicated by error bars.

### Burning results

Figure 4 shows chromatograms from I/SVOCs in the gas and particle phase from burning a
cow dung cake sample collected from SPE disks and PTFE filters during a whole 30-minute
burn, after passing through a dilution and cooling chamber. The saturation concentration $C_i^*$ at
298 K is provided as an alternative x-axis and has been calculated for each *n*-alkane, *i*, using:

$$C_i^* = \frac{M_i 10^6 \zeta_i P_{L,i}^o}{760RT} \qquad\qquad \text{E1}$$


where $M_i$ = molecular weight of VOC *i* (g mol$^{-1}$), $\zeta_i$ = activity coefficient of VOC *i* in the
condensed phase (assumed to be 1), $P_{L,i}^o$ = liquid vapour pressure of VOC *i* in Torr, *R* = gas





constant ($8.206 \times 10^{-5}$ m$^3$ atm mol$^{-1}$ K$^{-1}$) and $T$ = temperature in Kelvin (Lu et al., 2018). The
constant 760 Torr has been used to convert between units of atm and Torr where 1 atm = 760
Torr. $P^{o}_{L,i}$ values have been calculated from EPA Estimation Programme Interface Suite data at
298 K (EPA, 2012). The SPE disks showed 1295 peaks with unique mass spectra and captured
gaseous VOCs and I/VOCs with $C^* \sim 1\times10^8$ - $5\times10^2$ µg m$^{-3}$ at 298 K. The largest peaks were
from alkanes, 1-alkenes, limonene, phenolics, substituted naphthalenes, furans and substituted
pyridines. The PTFE filters captured 1604 I/SVOCs and low/non-volatility VOCs (L/NVOC)
with unique mass spectra present in the aerosol phase from $C^* \sim 5\times10^6$-$1\times10^{-5}$ µg m$^{-3}$ at 298
K. A transition can be seen in the two chromatograms from the gas to the aerosol phase. Species
with a saturation vapour concentration less than $5\times10^4$ µg m$^{-3}$ at 298 K were predominantly in
the aerosol phase after passing though the dilution chamber. A large region of more polar
components was present in the I/SVOC region from $C^*$ $5\times10^4$-$5\times10^0$ µg m$^{-3}$ at 298 K and
contained sugar pyrolysis products and highly substituted aromatics such as those with ketone,
ether and di and trisubstituted phenol substituents. Many alkanes, from *n*-octadecane to *n*-
triatriacontane were present, mainly in the SVOC region. The LVOC region was dominated by
a series of sterols and stanols. GCxGC provided extremely high resolution to allow
deconvolution of complex samples. The insert in Figure 4 shows how the complexity of the
SPE chromatogram can be further resolved by looking at a single ion chromatogram, for
example *m/z* = 57, which highlighted aliphatic non-polar peaks, with large peaks for alkanes
from *n*-nonane to *n*-nonadecane.
Figure 5 shows overlaid peak markers from SPE disks and PTFE filters from a 30-minute cow
dung cake burn coloured by functionality and phase. Over 3000 peaks with individual mass
spectra were identified. The complexity of emissions was vast, with 473 PAHs (light brown)
forming a group towards the top centre to right of the chromatogram. The most abundant
calibrated PAH in the gas phase was naphthalene, followed by methyl and dimethyl
naphthalene isomers. A range of methyl, dimethyl, tri and tetramethyl naphthalenes as well as
ethyl, propyl, butyl and methyl propyl isomers were detected. Naphthalene isomers substituted
with aldehydes, carboxylic acids and nitriles were also released. Biphenyl and a range of
methyl, dimethyl and ethyl biphenyls were also released. A range of other PAHs such as
acenaphthylene, fluorene, azulene, quinoline, chamazulene, benzophenone, stilbene and
benzofurans along with their alkyl substituted isomers were also in the gas phase. A large
amount of highly substituted, larger PAHs with more than 3 aromatics rings in their structure
were present in the aerosol phase.



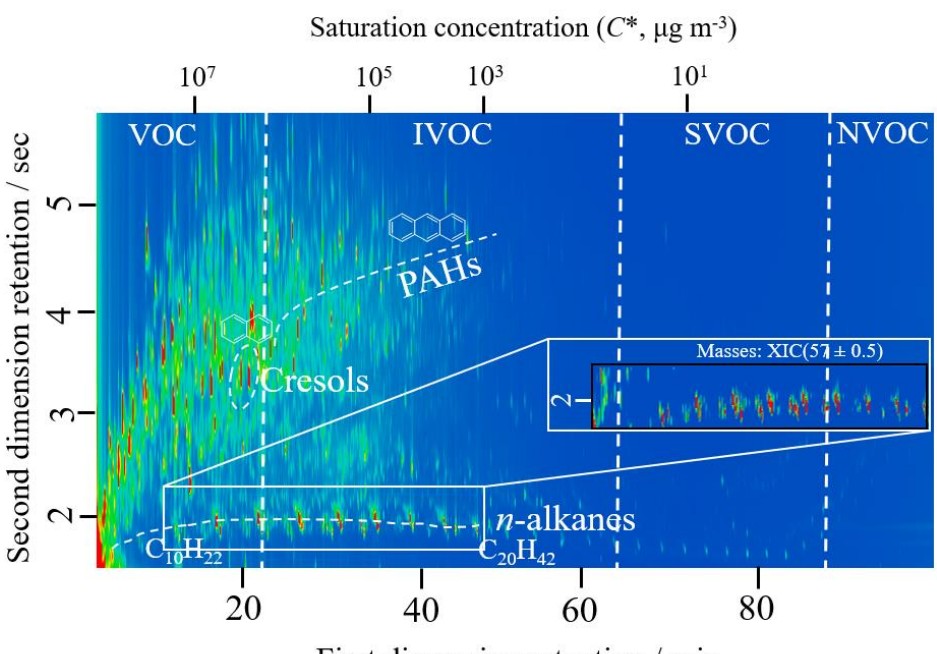

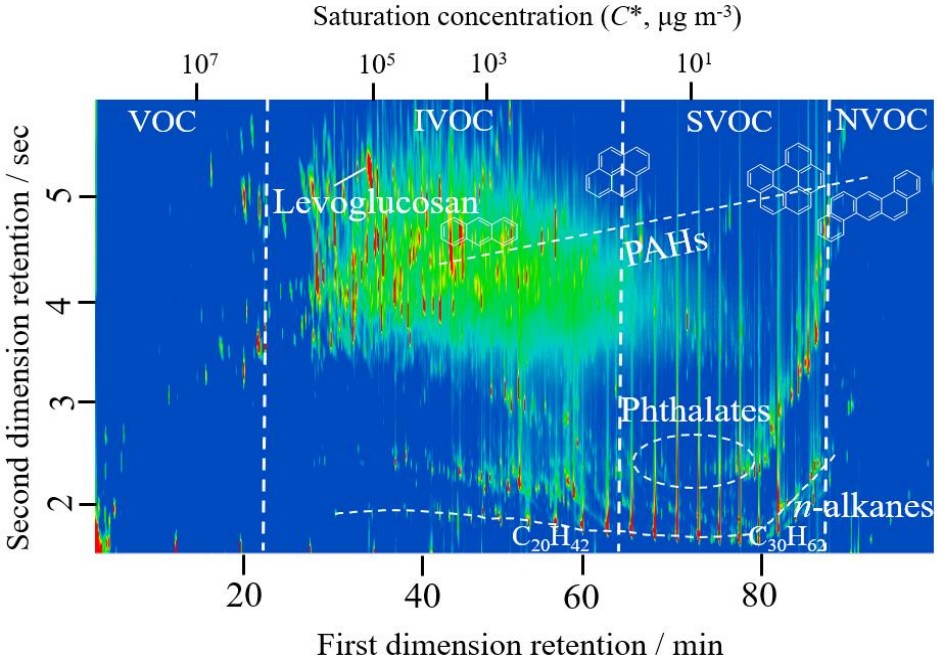

Figure 4. Chromatogram of SPE (top) and PTFE (bottom) extracted samples from the entire
burn of cow dung cake. *n*-Alkane and PAH series are marked on the chromatograms. The
saturation concentration scale matches the *n*-alkane series.

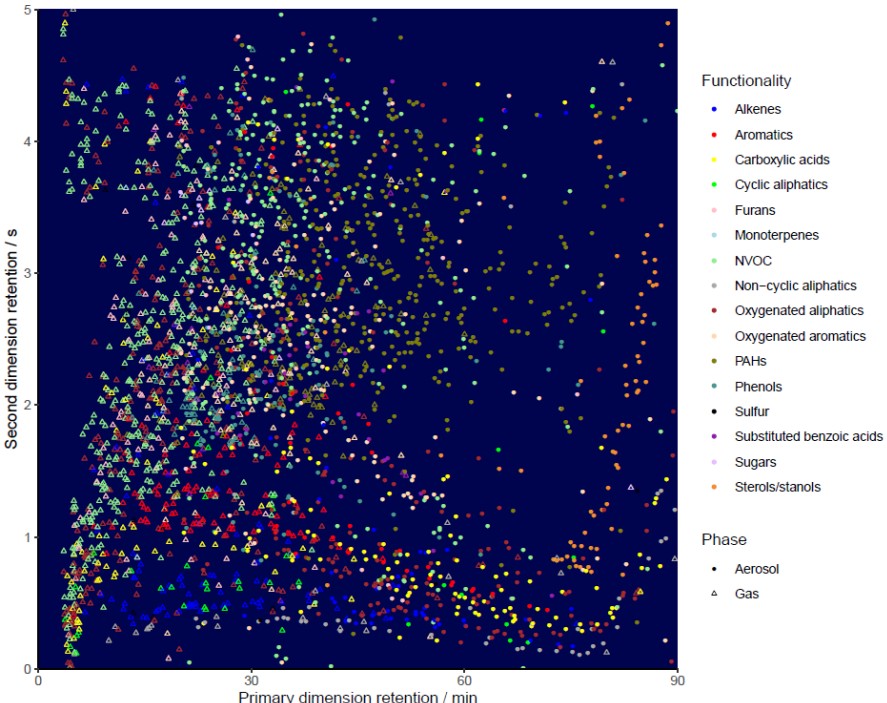


Figure 5. Gas and particle phase composition of I/SVOC emissions from burning cow dung cake collected onto SPE disks and PTFE filters, split by functionality where empty triangles indicate peaks in the gas phase and solid circles show peaks in the aerosol phase.


Other peaks present in Figure 5 included 145 alkenes, mostly towards the bottom of the
chromatogram, along with a row of 95 non-cyclic aliphatic and 44 cyclic aliphatic species.
Above was a row of 406 substituted aromatics, 208 carboxylic acids and 79 sterols/stanols as
well as 753 oxygenated hydrocarbons containing a range of ether, alcohol and aldehyde
functionalities. 250 peaks were from oxygenated aromatics, 170 from phenols and 44 from
substituted benzoic acids. In addition to this there were 118 furanic species, 3 monoterpenes
and 161 sulphur containing VOCs.
A wide array of NVOCs were present in the cow dung cake samples, with over 600 nitrogen
containing peaks including aromatics such as pyridines and pyrizines (123), amines (82),
amides (77), nitriles (74), 7-membered heterocycles (1), 6-membered heterocycles (28), 5-
membered heterocycles including aromatics such as pyroles as well as pyrolines and
pyrolidines (97), 4-membered heterocycles (6), 3-membered heterocycles (6), nitrogen
containing PAHs (38) imidazoles (22), imines (4), isocyanates (3), hydrazines (7), carbamic



acids (3), azoles (33) oximines (3) and sulfur containing nitrogen compounds (14). Previous
studies have measured the nitrogen content of cow dung cake to be as high as 1.9 % (Stockwell
et al., 2014) in comparison to other fuel types such as fuel woods (0.14-0.35 %), rice straws
(0.4 %) and coal (0.6 %). The large amount of NVOCs are likely formed from the volatilisation
and decomposition of nitrogen-containing compounds within the cow dung cake, such as free
amino acids, pyrroline, pyridine and chlorophyll (Leppalahti and Koljonen, 1995; Burling et
al., 2010; Ren and Zhao, 2015). NVOCs are of concern because they can be extremely toxic
(Ozel et al., 2010b; Ozel et al., 2010a; Ozel et al., 2011; Ramírez et al., 2012, 2014; Ramírez
et al., 2015; Farren et al., 2015) and amines in particular can change the hydrological cycle by
leading to the creation of new particles (Smith et al., 2008; Kirkby et al., 2011; Yu and Luo,
2014) which act as cloud condensation nuclei (Kerminen et al., 2005; Laaksonen et al., 2005;
Sotiropoulou et al., 2006).
Figure 6 shows a comparison of organic aerosol composition observed from different fuel types
(LPG, fuel wood, sawdust and municipal solid waste). The measured emissions had very
different compositions, reflecting the variability of organic components produced from
different sample types. Sawdust, municipal solid waste and cow dung cake (shown in Figure
4) emitted a wide range and complexity of species. Particle phase emissions from LPG burning
were minimal, with most peaks from the internal standard or contaminants in the solvent. Fuel
wood samples released more organic components into the aerosol phase, with the majority of
IVOCs with $C^* \sim 1.2 \times 10^5 - 7 \times 10^1$ µg m$^{-3}$ at 298 K. The largest peak belonged to levoglucosan,
with other peaks from monoaromatics with several polar substituents such as ethers and
phenols, for example dimethoxyhydrotoluene and syringyl acetone. These were likely from the
depolymerisation of lignin (Simoneit et al., 1993; Sekimoto et al., 2018), an amorphous
polymer constituting about 25 % of fuel woods (Sjöström, 1993) and formed of randomly
linked, high-molecular weight phenolic compounds (Shafizadeh, 1982).
Sawdust, not a widely used fuel source, released many I/S/L/NVOC components in the aerosol
phase over a much wider range ($C^* \sim 5.8 \times 10^5 - 1 \times 10^{-3}$ µg m$^{-3}$ at 298 K). The largest peak was
from levoglucosan with another large peak from squalene. Many peaks were from polar
substituted aromatics as well as many PAHs and their substituents, such as 2-methyl-9,10-
anthracenedione. The largest peak from municipal solid waste burning was also levoglucosan,
but these samples released fewer of the polar substituted monoaromatics than other samples.
Municipal solid waste released alkanes and SVOC species such as terphenyls, alkanes and
many PAHs.

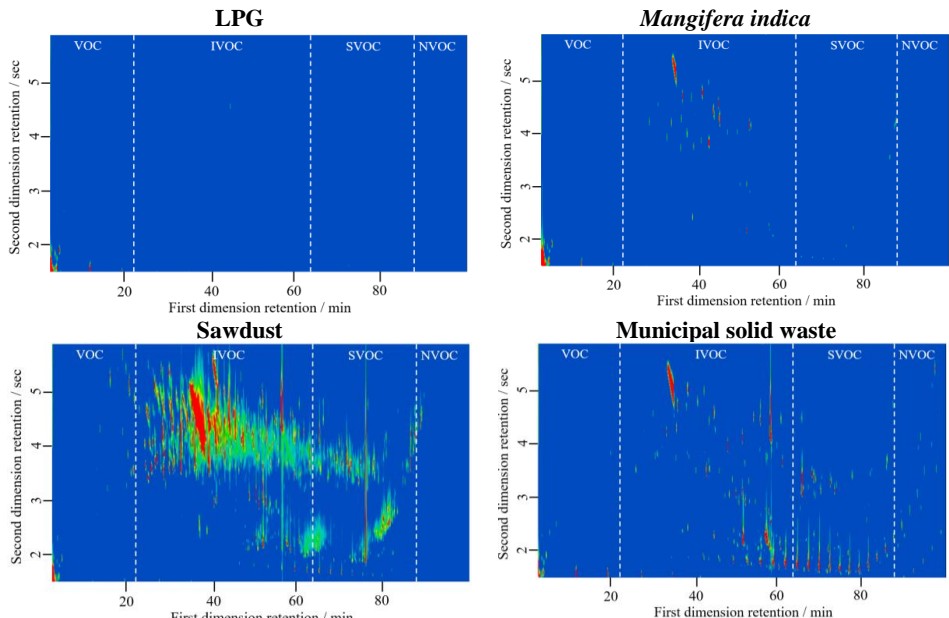

Figure 6. Measurements of organic aerosol from a range of different fuel types, with the contrast at the same scale.

**Molecular markers for domestic fuels**

Cow dung cake samples emitted a range of sterols/stanols, which have been reported previously (Sheesley et al., 2003). This study suggested that 5β-stigmastanol, coprostanol, and cholestanol could be used as tracers for emissions from cow dung cake burning. This is because in higher animals, anaerobic microbial reduction of sitosterol and cholesterol forms the distinctive β configuration of the C-5 proton of 5β-stigmastanol and coprostanol. This contrasts with the α C-5 proton caused by aerobic digestion in aquatic environments. Jayarathne et al. (2018) reported 5β-stigmastanol emissions from hardwood, and Fine et al. (2001) reported 5α-stigmastanol emissions from hardwood. Four fuel wood samples in our study showed emissions of an isomer of stigmastanol, a result similar to Jayarathne et al. (2018) that 5β-stigmastanol was not unique to cow dung cake burning or the MS measurement method used was unable to distinguish between 5α- and 5β-stigmastanol. Cholestanol and coprostanol were found uniquely in cow dung cake samples in our study, and suggested that they can be used as unique tracers for cow dung cake burning.

Fuel wood samples generally released fewer organic components into the aerosol phase than samples such as cow dung cake, MSW and sawdust. Levoglucosan has been traditionally suggested as a tracer for biomass burning emissions, however, emissions of levoglucosan from



a range of sources mean that this is of limited use as a unique tracer of woodsmoke emissions
in regions with multiple burning sources. This could potentially be resolved in future studies
by examining the ratio of levoglucosan to other sugars in different source types to differentiate
different biomass burning sources as the chemical composition of different sources should
determine the emission ratio of levoglucosan to other sugar pyrolysis products (Sheesley et al.,

452  2003).

The presence of a wide range of terphenyls in municipal solid waste samples in this study was
not unique. Jayarathne et al. (2018) suggested triphenyl benzene to be a unique tracer of waste
burning emissions. Whilst this study found triphenyl benzene present in one cow dung cake
sample and in municipal solid waste samples, the waste samples emitted on average 19
terphenyls, many more than in the cow dung cake samples (2). Terphenyls have been
previously reported from incineration of waste (Tong et al., 1984) and our study suggests that
these compounds are good indicators of municipal solid waste burning.
**Total identification**
Figures 7A and 7B show a comparison of the relative abundance of peaks identified, defined
here as the sum of peak areas identified and calibrated using genuine standards for compounds
present in the SPE and PTFE samples compared to the total observed peak area (using the blank
subtracted total ion current, TIC).
Figure 7A shows that between 15 and 100 % of the peak area of the TIC in the SPE
chromatogram could be identified. The highest proportion of species that could be identified
was from fuel wood (67 %), followed by crop residue (57 %), charcoal (48 %), municipal solid
waste (46 %), cow dung cake (39 %) and sawdust (16 %). Lower total identification in samples
such as cow dung cake was due to increased complexity of emissions, which were not covered
by the standards used.
Figure 7B shows that between 7 – 100 % of the organic composition of aerosol released from
burning was identified. Generally, a much lower proportion of organic matter within aerosol
samples was identified due to a lack of genuine standards available, particularly in complex
samples. The lowest mean relative contribution identified from samples was sawdust (9 %),
followed by cow dung cake (11 %) and municipal solid waste (16 %). A larger relative
contribution was identified from fuel woods (34 %) and charcoal (39 %) and due to less
complex emissions. A large relative contribution of some fuel woods was identified from
*Saraca indica* (91 %) and *Pithecellobium spp* (82 %) due to a low amount of organic matter



released from these samples. This also influenced the percentage identification from crop
residue which achieved 46 % identification, due to only 3 samples with 98 % identification
from *Solanum melongena* but only 26 % from *Cocos nucifera* and 13 % from *Brassica spp*.
100 % of the aerosol released from LPG was quantified due to little being released into the
aerosol phase and this was principally composed of PAHs. These low levels of identification
of organic aerosol are in line with those reported by Jen et al. (2019) where unknown chemical
species represented 35-90 % of the observed organic aerosol mass from biomass burning
samples.
**Composition**
Figure 7C provides an indication of I/SVOC composition on SPE disks by mass of quantified
species, assuming no compound breakthrough. Phenolic and furanic compounds are the most
abundant I/SVOC species released from all sample types, except for LPG. As a proportion of
the total mass of species quantified with genuine standards on SPE disks, phenols released from
fuel woods (22-80 %) represented the largest range, with large amounts released from
municipal solid waste (24-37 %), cow dung cake (32-36 %), crop residue (32-57 %) and
sawdust (46 %). High emissions of phenolic compounds were of significance because
phenolics contribute significantly to SOA production from biomass-burning emissions (Yee et
al., 2013; Lauraguais et al., 2014; Gilman et al., 2015; Finewax et al., 2018).
Large emissions of furanic species were measured from fuel wood (6-59 %), municipal solid
waste (35-45 %), cow dung cake (39-42 %), crop residue (25-44 %) and sawdust (43 %). These
were important as furans can be toxic and mutagenic (Ravindranath et al., 1984; Peterson,
2006; Monien et al., 2011; WHO, 2016) and have been shown to be some of the species with
the highest OH reactivity from biomass burning emissions (Hartikainen et al., 2018; Coggon
et al., 2019). Furans have also been shown to result in SOA production (Gómez Alvarez et al.,
2009; Strollo and Ziemann, 2013) with 8-15 % of SOA produced from combustion of black
spruce, cut grass, Indonesian peat and ponderosa pine estimated to originate from furans and
28-50 % of SOA from rice straw and wiregrass (Hatch et al., 2015). Furans from biomass
burning emissions are thought to come from low temperature depolymerisation of hemi-
cellulose (Sekimoto et al., 2018) and from large alcohols and enols in high-temperature regions
of hydrocarbon flames (Johansson et al., 2016).
Emissions of alkanes were most important from combustion of cow dung cake and municipal
solid waste (4-9 %), with only small quantities released from various fuel wood samples (< 2



%) and crop residue (< 1%). This reinforced previous studies which found emissions of $C_{12}$-
$C_{39}$ *n*-alkanes from municipal waste incinerators (Karasek and Tong, 1985). PAH emissions
represented (3 – 15 %) of the total quantified emission by mass for samples other than LPG
and have carcinogenic and mutagenic properties (IARC, 1983, 1984; Nisbet and LaGoy, 1992;
Lewtas, 2007; Zhang and Tao, 2009; Jia et al., 2011). They can damage cells through the
formation of adducts with DNA in many organs such as the kidneys, liver and lungs (Vineis
and Husgafvel-Pursiainen, 2005; Xue and Warshawsky, 2005).
Figure 7D shows the quantified aerosol mass was largely dominated by levoglucosan, with a
particularly significant contribution in the fuel wood samples (13-98 %). This was similar to a
previous study of fuel wood samples from Bangladesh, where levoglucosan was the largest
contributor to aerosol mass (Sheesley et al., 2003). Levoglucosan emissions were also large
from cow dung cake (30-58 %), which contrasted with the findings of Sheesley et al. (2003).
This could be due to differences in the feeding of cows leading to differences in residual
undigested organic matter in cow dung cake samples as well as differences in preparation
between samples collected in Bangladesh and those in this study, which had additional dried
biogenic material, such as straw, mixed into samples. Levoglucosan emissions were also high
from sawdust (91 %), crop residue (19-85 %) and municipal solid waste (58-75 %), with
municipal solid waste emissions likely from cellulosic material collected with samples.
Levoglucosan emissions from charcoal (76 %) were significant as a proportion of emissions.
Emissions from charcoal were low, which meant that a small emission of levoglucosan
represented a large proportion of total emissions. It was likely that the sample collected here
may have contained small amounts of cellulosic organic matter that led to the emission of
levoglucosan.
Emissions of alkanes in the gas and particle phases were similar by source type, with particulate
alkanes emitted principally from cow dung cake and municipal solid waste samples. Emissions
of particulate phenolics were large as a proportion of total quantified mass with genuine
standards when the total emission of other components was low. For example, phenolics
represented a large proportion of emissions from the fuel wood species *Morus spp* and
*Pithecellobium spp* with the mass principally from dimethoxyphenols. Emissions from LPG
were mainly PAHs and very low.
Whilst SPE samples for these compounds remained semi-quantitative due to slight
breakthrough, the detection of high emissions of phenolics and furanics in the gas phase from





burning was in line with recently published studies (Hatch et al., 2015; Stockwell et al., 2015;
Koss et al., 2018). Relatively low levels of total quantified material within the aerosol phase
was in line with the current literature (Jen et al., 2019) but meant that this analysis was not
entirely reflective of the organic fraction for complex samples. It is likely that this study
overemphasises the contribution of levoglucosan in complex aerosol samples, relative to other
components present at lower levels (Sheesley et al., 2003; Jen et al., 2019). Future instrument
development could allow better quantification of complex burning and ambient samples by
splitting the eluent between a -MS and -FID. This study suggests that future research uses lower
sample volumes, thicker SPE disks and studies the adsorption characteristics of VOCs to the
surfaces of these disks.

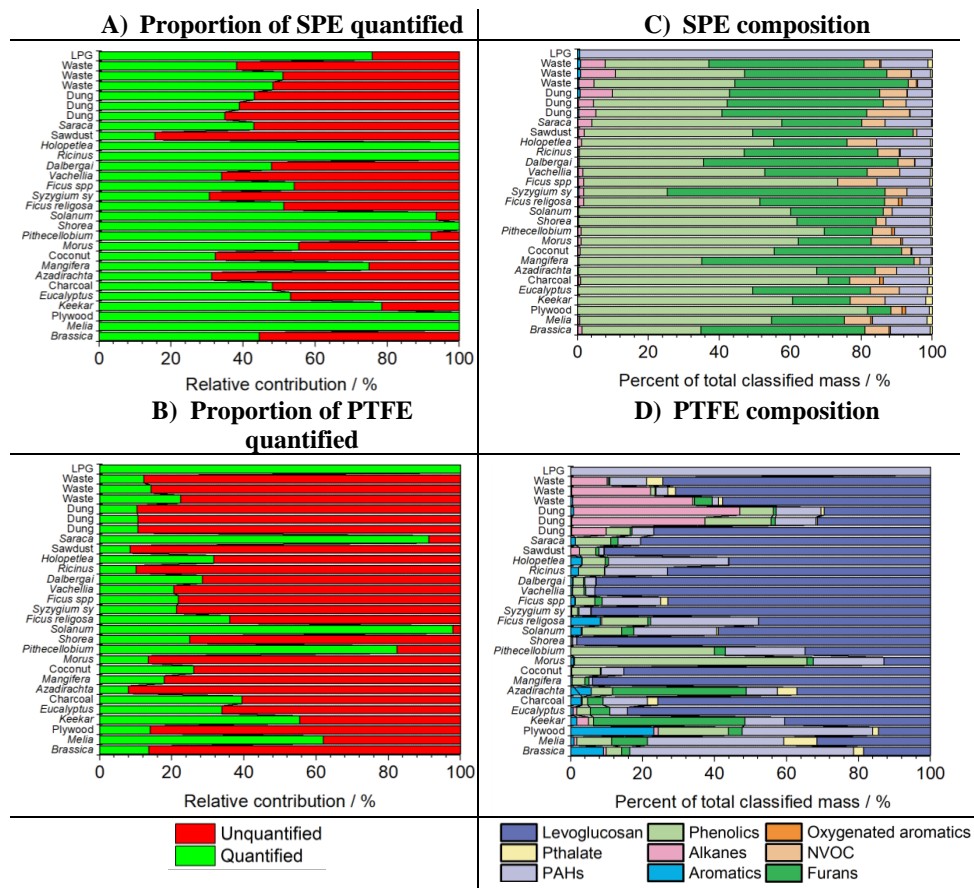

Figure 7. Area of organic matter quantified with genuine standards, as a fraction of total ion current
(TIC) (7A and 7B, left panel). Semi-quantitative/quantitative analysis of SPE/PTFE fraction (7C and
7D, right panel).





**Development of emission factors**

Emission factors have been developed for PAHs (see Figure 8 and the Supplementary Information S5 for table of emission factors by individual fuel type) by calculating the total volume of air convectively drawn up the flue and relating this to the mass of fuel burnt (see the Supplementary Information S6 for details of calculation). Emission factors for sawdust (1240 mg kg$^{-1}$), municipal solid waste (1020 mg kg$^{-1}$), crop residue (747 mg kg$^{-1}$) and cow dung cake (615 mg kg$^{-1}$) were generally larger than for fuel wood (247 mg kg$^{-1}$), charcoal (151 mg kg$^{-1}$) and LPG (56 mg kg$^{-1}$). The measurement of higher emission factors for cow dung cake than fuel wood was consistent with that observed in other studies (Bhargava et al., 2004; Gadi et al., 2012).

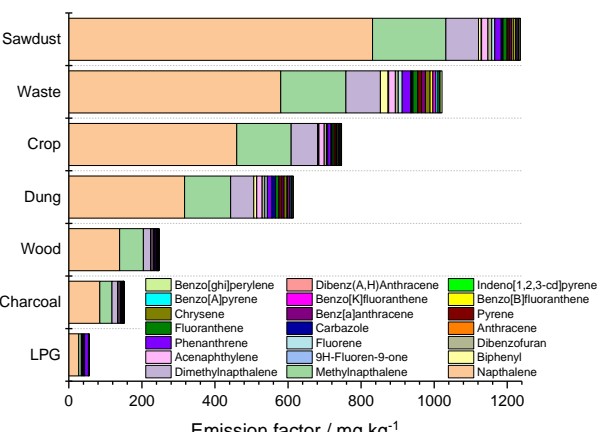

Figure 8. Mean PAH emission factors by fuel type.

A wide range of emission factors were measured for fuel wood samples from 50 mg kg$^{-1}$ for *Prosopis* to 907 mg kg$^{-1}$ for *Ficus religosa*. For most samples, PAH emissions in the gas phase were dominated by naphthalene, methylnaphthalenes and dimethylnaphthalenes with gas-phase PAHs observed up to pyrene. For fuel wood, crop residue, municipal solid waste and cow dung cake the percentage of PAHs in the gas phase decreased from 97 %, 96 %, 91 % to 89 %. PAHs from LPG showed the largest fraction in the gas phase (99.9 %) compared to the aerosol phase (0.1 %). Figure 9 shows gas and particle phase PAH emissions by individual sample type, excluding naphthalene as well as $C_1$- and $C_2$-substituted naphthalenes. PAHs were present in the aerosol phase from dibenzofuran ($C_{12}H_8O$) to benzo(ghi)perylene ($C_{22}H_{12}$).

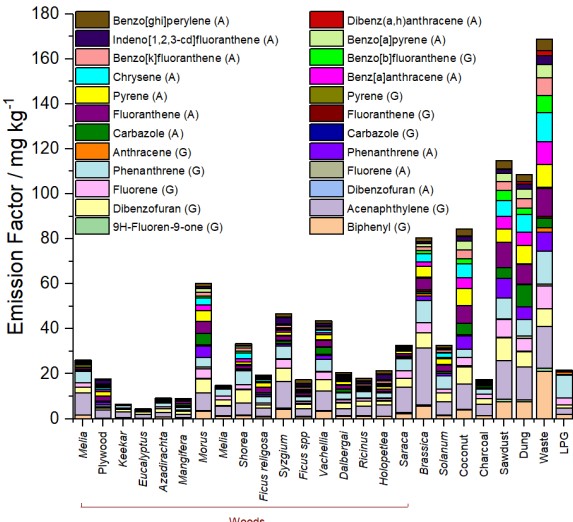

578

Figure 9. Emission factors of PAHs measured from SPE/PTFE where (G) and (A) represent gas- and aerosol-phase samples, respectively, excluding naphthalene as well as naphthalenes with $C_1$ and $C_2$ substituents.

Table 3 shows a comparison of the mean emission factors measured in our study with previous studies. The mean fuel wood total PAH emission factor measured in our study (247 mg kg$^{-1}$) was a factor 4.7-5.6 larger than those measured by Gadi et al. (2012) and Singh et al. (2013) of 44 and 53 mg kg$^{-1}$, respectively, for similar fuel woods collected across New Delhi and the Indo-Gangetic Plain. The PAH emission factor measured for cow dung cake (615 mg kg$^{-1}$) was around a factor of 10 larger than those previously measured (60 mg kg$^{-1}$). The larger total emission factors for fuel wood and cow dung cake was a result of high emissions of gas-phase PAHs measured using PTR-ToF-MS (51-896 mg kg$^{-1}$ for fuel wood and 446-660 mg kg$^{-1}$ for cow dung cake) compared with previous measurements made using PUF plugs (7 mg kg$^{-1}$). This indicated that either the PTR-ToF-MS was able to better detect and characterise gas-phase emissions than previous methods and suggested either breakthrough or off gassing of smaller gas-phase PAHs from PUF plugs or measurement of significant quantities of other $C_{10}H_8$ isomers on the PTR-ToF-MS. This may highlight an underestimation of 2-ring gas-phase PAH emissions in previous burning studies. Gadi et al. (2012) measured PAH emissions in the particle phase, with the mean emission for fuel wood (44 mg kg$^{-1}$) greater than our study (9 mg kg$^{-1}$). Particulate phase emissions of PAHs measured by Singh et al. (2013) from fuel wood (45 mg kg$^{-1}$) were also larger than our study. By contrast, particle phase PAH emissions from cow dung cake in our study (66 mg kg$^{-1}$) were comparable to those measured previously of 57-60 mg kg$^{-1}$ (Gadi et al., 2012; Singh et al., 2013). Variability in emission of particulate-phase





PAHs in our study compared to literature was likely to be highly influenced by the efficiency
of combustion of samples. Although not measured in our study, differences in moisture content
between samples in our study and literature were likely have a large influence on the total
amount of PAHs emitted and may explain the differences in particle-phase emissions.
The particulate phase PAH emission factors from municipal solid waste combustion in our
study (14-181 mg kg$^{-1}$) were much smaller than those of previous studies (1910-8486 mg kg$^{-1}$
$^{-1}$), but the number of samples was limited. Emissions from coconut shell have not been well
studied, making comparisons difficult (Gulyurtlu et al., 2003). The emission of particulate
phase PAHs from sawdust in our study (62 mg kg$^{-1}$) was less than that previously reported 259
mg kg$^{-1}$, but our study found large gas phase PAH emissions (1175 mg kg$^{-1}$). Particulate PAH
emissions from the crop residue burnt in our study (13-53 mg kg$^{-1}$) fell within the range
reported by Kim Oanh et al. (2015) of 0.34-34 mg kg$^{-1}$ for rice straw. Those reported by Wiriya
et al. (2016) were smaller (0.47 mg kg$^{-1}$), but were from samples dried in an oven at 80 ºC for
24 hours and ignited by an LPG burner and were likely to represent more complete combustion
conditions. Emissions of PAHs from charcoal in our study (151 mg kg$^{-1}$) were larger than those
measured for South Asian fuels (25 mg kg$^{-1}$), caused principally by larger measurement of gas-
phase species by PTR-ToF-MS. Both our study, and that of Kim Onah et al. (1999) showed
charcoal released the least amount of PAH per kg burnt for biofuels. LPG combustion released
less particulate PAHs (0.1 mg kg$^{-1}$) than previous studies (0.8 mg kg$^{-1}$), but also included a
small gas-phase emission (56 mg kg$^{-1}$). Differences in the distribution of PAHs found in the
gas and aerosol phases between our study and literature were also likely to be influenced by
the different sample dilutions and gas-to-aerosol partitioning prior to measurement.
Table 3. PAH emission factors measured in our study compared to values from literature for similar
fuel types.

| Fuel | PAH (mg kg$^{-1}$) | | | |
|------|------|------|------|------|
| | **Gas** | **Particle** | **Total** | **Ref** |
| **Wood** | 51-896 | 0.4-34 | 51-907 | Our study |
| | | 1-12 | | (Hosseini et al., 2013) |
| | 22-111 | 0.4-6 | 24-114 | (Kim Oanh et al., 2005) |
| | - | 44 | 44 | (Gadi et al., 2012) |
| | 7 | 45 | 52 | (Singh et al., 2013) |
| | | 805-7294 | | (Kakareka et al., 2005) |
| | | | 43 | (Lee et al., 2005) |
| | 66 | 0.8 | 67 | (Kim Oanh et al., 2002) |
| | 105 | 4 | 105 | (Kim Oanh et al., 1999) |
| **Dung** | 446-660 | 48-98 | 493-710 | Our study |
| | - | 59 | - | (Gadi et al., 2012) |
| | 3 | 57 | 60 | (Singh et al., 2013) |




| Waste | 696-1233 | 14-181 | 776-1414 | Our study |
|---|---|---|---|---|
| | - | 8486 | 8486 | (Kakareka et al., 2005) |
| | - | 1910 | 1910 | (Young Koo et al., 2013) |
| Crop | 205-1231 | 13-53 | 219-1255 | Our study |
| | - | - | 5-683 | (Jenkins et al., 1996) |
| | - | - | 3-50 | (Lu et al., 2009) |
| | - | - | 129-569 | (Wei et al., 2014) |
| | 5-230 | 0.3-34 | 5-264 | (Kim Oanh et al., 2015) |
| | - | 0.47 | - | (Wiriya et al., 2016) |
| Sawdust | 1175 | 62 | 1236 | Our study |
| | 259 | 261 | | (Kim Oanh et al., 2002) |
| Charcoal | 147 | 4 | 151 | Our study |
| | 25 | 0.1 | 25 | (Kim Oanh et al., 1999) |
| LPG | 56 | 0.1 | 56 | Our study |
| | - | 0.8 | - | (Geng et al., 2014) |


**Conclusions**

This paper demonstrated an extraction technique for biomass burning samples collected onto SPE disks and PTFE filters, which was well suited to analysis of non-polar species. A range of samples relevant to burning in India were collected and analysed, which showed large differences in the composition of organic matter released. The separation power of GCxGC has been used to identify an extensive range of I/SVOCs in both gas and particle phases with 15-100 % of gas-phase emissions and 7-100 % of particle-phase emissions characterised.

The ability to quantify species on SPE disks was assessed and scope for future studies which should assess the adsorption characteristics of IVOCs onto SPE disks has been provided. It is recommended that breakthrough of IVOCs collected onto SPE disks at lower sample volumes is evaluated, and better methods for quantification of complex samples are developed. Further samples from a wider range of sources would enable a better understanding of the drivers of poor air quality in the developing world, such as crop residue burning. This study found that cholestanol and coprostanol were unique to cow dung cake burning samples and these species were therefore suggested as tracers for emissions from cow dung cake burning. Similarly, municipal solid waste burning released many terphenyls, which could act as good indicators of this source. This study found that phenolic and furanic species were the most important gas-phase emissions by mass of I/SVOCs from biomass burning. New emission factors were developed for US EPA criteria PAHs present in gas and aerosol phases from a large range of fuel types. This suggested that many sources important to air quality in the developing world are larger sources of PAHs than conventional fuel wood burning.



*Author contributions.* GJS developed the ASE method, GC method, collected samples,
organised logistics, extracted/analysed samples and lead the paper. BSN collected samples and
assisted with logistics. WJFA measured VOCs by PTR-ToF-MS, supported by CNH, LKS and
NT. ARV assisted in running and organising of experiments. NJF, JRH and MWW assisted in
GCxGC-ToF-MS method development. SJS assisted in ASE method development. RA, AM,
RJ, SA and LY collected samples, carried out the burning experiments and measured gas
volumes up the flue. SSBMY aided complex sample analysis. EN, NM, RG, SKS and JDL
contributed to logistics and data interpretation. TKM and JFH provided overall guidance with
setup, conducting, running and interpreting experiments.
*Competing interests.* The authors declare that they have no conflict of interest.
*Acknowledgements.* This work was supported by the Newton-Bhabha fund administered by the
UK Natural Environment Research Council, through the DelhiFlux project of the Atmospheric
Pollution and Human Health in an Indian Megacity (APHH-India) programme. The authors
gratefully acknowledge the financial support provided by the UK Natural Environment
Research Council and the Earth System Science Organization, Ministry of Earth Sciences,
Government of India under the Indo-UK Joint Collaboration vide grant nos NE/P016502/1 and
MoES/16/19/2017/APHH (DelhiFlux) to conduct this research. The paper does not discuss
policy issues and the conclusions drawn in the paper are based on interpretation of results by
the authors and in no way reflect the viewpoint of the funding agencies. GJS and BSN
acknowledge the NERC SPHERES doctoral training programme for studentships. RA, AM,
RJ, SA, LY, SKS and TKM are thankful to Director, CSIR-National Physical Laboratory, New
Delhi for allowing to carry out this work. LKS acknowledges Physical Research Laboratory
(PRL), Ahmedabad, India for the support and permission to deploy PTR-ToF-MS during the
experimental campaign. All authors contributed to the discussion, writing and editing of the
manuscript.



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
