# Peer review of "domestic fuels used in Delhi, India"

_Atmospheric Chemistry and Physics, 2020_

## Referee Comment (RC1) · Anonymous Referee #1 · 7 Sep 2020

Stewart et al present an extremely detailed analysis of emissions of lower-volatility vapors (I/SVOCs) and particulate matter from domestic fuel burning. They focused on fuels typically burned in India. One novel aspect of this study is the use of GCxGC to identify hundreds of individual species emitted from each fuel type.

Overall the paper is very well written and easy to follow. The description of the methods, in particular, is very detailed and shows the high level of specificity in the measurements and the extensive QA/QC. The paper is a little bit tough to follow because the section (and sub-section) headers all have the same style and font. This makes it hard to determine the organization of the manuscript (e.g., there are several sub-sections to

the Methods, but the headers are indistinguishable from the following Results section). Consider numbering the sections.

This paper presents a potentially valuable dataset containing measurements of hundreds of species. However, only the emission factors for PAHs are listed in detail in Table 1 of the SI. I think it would be beneficial for the authors to share the full suite of measured compounds, either in the SI or in an online repository, so that data can be used by other researchers in the future.

One potential weakness is that there is only one sample for most of the fuels tested (Table 1). The authors should acknowledge that there can be significant burn-to-burn differences in emissions.

I'm not certain that Table 2 should be in the main text. This table seems to be part of the QA/QC, and it seems to me that it would be better placed in the SI.

Figure 5 is hard to interpret. The symbols are very small (as is the legend), and as the authors note in the text, there are a lot of species shown. Since the text focuses on the PAHs, it might help readability to put this version of the figure in the SI and only show the PAHs (with larger symbols) in the main text.

Figure 7 and line 548-550 suggest that more of the mass could be speciated with new instruments. However Figure 7 focuses on peaks that can be positively matched with something in one of the standards. Is it possible to infer composition based on the mass spectra of the unidentified peaks?

Grammatical comments: Line 152 and 153 - it seems like "samples" in the former line mean the sampled media (filters and SPE), and in the latter it means the fuels. Please clarify.

Line 195 - define EtOAc

Line 347 uses NVOC to indicate "non-volatile", but the paragraph starting at line 389 seems to use NVOC to indicate "nitrogen-containing." Please clarify.

---

## Referee Comment (RC2) · Anonymous Referee #2 · 21 Oct 2020

Stewart et al. have developed a gas-chromatography-based analytical technique to speciate and quantify semi-volatile and intermediate volatility organic compounds (S/IVOCs) and applied this technique to measure emissions of S/IVOCs from domestic fuels used in Delhi, India. S/IVOCs are important precursors to ozone and aerosol formation in the atmosphere and there is need to develop robust analytical techniques to speciate and quantify their emissions. Biomass burning is an important source of global air pollution and the type of biomass burning studied here (i.e., biofuel combustion) is a particularly understudied emission source. Hence, the work described in the manuscript is well motivated. I should also commend the authors for a well written manuscript that provides all the necessary details to comment on the methods and the

interpretation of the results. The analytical method development was well designed and the application was very well described, although I should note that I am not trained as an analytical chemist. This should serve as a useful resource for researchers doing similar work in the atmospheric community. The primary results of speciation and quantification are well described too but, given the large dataset that is being analyzed, only a small fraction of the data are actually presented. I recommend the publication of this study in Atmospheric Chemistry and Physics after the authors have had a chance to respond to my, mostly big-picture and minor, comments.

Big-picture comments: 1. The introduction seems too generic and long at the moment and needs to be realigned to describe the state-of-the-science and gaps as it relates to the key findings from this work. For instance, lines 53-68 discuss S/IVOC emissions generally but don't focus on those emissions from biomass burning. Text from lines 69 to 107 could be condensed into a few sentences. Earlier work relevant to this paper seems to be mentioned in lines 108 to 145 and needs to be highlighted, front and center. Another point that could be highlighted is that S/IVOC emissions are poorly, if at all, represented in emissions inventories and chemical transport models and their impacts on atmospheric chemistry and air quality are uncertain (with particular relevance to regions where this and similar fuel use is dominant, e.g., Asia). 2. Given the large variability seen in biomass burning emissions, say relative to internal combustion engines, the authors should comment on the single experiments done for most of the fuels. This could be done by analyzing the experiment-to-experiment variability for the fuels where multiple experiments were done (i.e., cow dung cake, waste), as well as through a review of similar literature. In addition, they should also comment on the differences in combustion encountered in their setup versus a real-world application. For example, most municipal solid waste is probably burned in a high-temperature incinerator where the combustion chemistry might be very different than the combustion simulated in this work. I do understand that 'backyard' low-temperature MSW fires are a major concern in India, including in Delhi. 3. I commend the authors on putting together this fantastic dataset of speciation and emission factors and I am fairly certain that this will serve as

a comprehensive resource for years to come (from studying exposure to toxic pollutants to developing accurate emissions inventories for air quality modeling). [This must be an oversight but I did not see a 'data availability' section that describes how and where the data will be archived for others to use]. However, the manuscript seems to present only a 'snapshot' of the dataset, with a mix of higher-level observations and depth for only a subset of speciated organic compounds (e.g., PAHs). Correct me if I am wrong but there is so much more to the dataset than what is presented. If that is indeed the case, what I would have liked to see is a structured vision for how the data plans to be analyzed further (e.g., detailed source profiles, molecular markers for source identification, volatility distributions) and what open, pressing questions would this dataset help answer in the long run?

Minor comments: 1. Line 228: Explain what 'NIST library hit was >800' means. 2. Line 471-485: Clarifying questions. Is the low fraction of the speciation of the organic aerosol limited to not finding a match in the NIST or does it highlight a problem with the analytical method? In addition, how sensitive is the fraction speciated to the use of the filter media, i.e., better with PTFE versus quartz? 3. The composition section could benefit from findings from some recent publications that have studied SOA from biomass burning emissions or precursors, e.g., He et al. (ESPI, 2020) – alkylfuran mixture, Joo et al., (ESC, 2019) – 3-methylfuran, Ahern et al. (JGR, 2019), Akherati et al. (ES&T, 2020), and Lim et al. (ACP, 2019) – biomass burning SOA in laboratory experiments with an emphasis on understanding phenolic, furanic, and monoterpene VOC contributions to SOA. 4. Figure 7: Was total organic mass in the gas- and particle-phase measured another way, e.g., FID-gas, Sunset OC/EC-particle, to get mass closure? 5. Figure 8: Mention sample size for each fuel. Specify measurement uncertainty when n=1. Combine measurement uncertainty and experiment-to-experiment variability when n>1. 6. Figure 9: Was the gas/particle partitioning of PAHs analyzed further? Seems like an ideal dataset to study absorptive partitioning. 7. Lines 631-632: Why were certain samples fully speciated and others not very much? Was this relationship examined further with respect to its sensitivity to fuel, total organic mass

captured on filter/disc, other variables?

---

## Author Response (AR1)

**Point-by-point response to reviewer comments**

We would like to thank the reviewers for their positive and constructive reviews of this paper. We address the specific points of each reviewer below. Reviewer comments in blue, author response in black, text added or amended in paper in purple.

Reviewer comment 1

Stewart et al present an extremely detailed analysis of emissions of lower-volatility vapours (I/SVOCs) and particulate matter from domestic fuel burning. They focused on fuels typically burned in India. One novel aspect of this study is the use of GCxGC to identify hundreds of individual species emitted from each fuel type.

Overall the paper is very well written and easy to follow. The description of the methods, in particular, is very detailed and shows the high level of specificity in the measurements and the extensive QA/QC. The paper is a little bit tough to follow because the section (and sub-section) headers all have the same style and font. This makes it hard to determine the organization of the manuscript (e.g., there are several sub-sections to C1 ACPD Interactive comment Printer-friendly version Discussion paper the Methods, but the headers are indistinguishable from the following Results section). Consider numbering the sections.

We thank the reviewer for their suggestion and all sections are now numbered.

This paper presents a potentially valuable dataset containing measurements of hundreds of species. However, only the emission factors for PAHs are listed in detail in Table 1 of the SI. I think it would be beneficial for the authors to share the full suite of measured compounds, either in the SI or in an online repository, so that data can be used by other researchers in the future.

We have added this to the SI.

One potential weakness is that there is only one sample for most of the fuels tested (Table 1). The authors should acknowledge that there can be significant burn-to-burn differences in emissions.

This is indeed a limitation and is now acknowledged in the main text.

Variability in emission of particulate-phase PAHs in our study compared to literature was likely
to be highly influenced by the efficiency of combustion of different fuel types. This may also
be explained by measuring only once for many of the fuel types, due to significant burn-to-
burn differences in emissions.

I'm not certain that Table 2 should be in the main text. This table seems to be part of the
QA/QC, and it seems to me that it would be better placed in the SI.

This table is now in the SI.

Figure 5 is hard to interpret. The symbols are very small (as is the legend), and as the authors
note in the text, there are a lot of species shown. Since the text focuses on the PAHs, it might
help readability to put this version of the figure in the SI and only show the PAHs (with larger
symbols) in the main text.

Figure 5 is now in the SI and this figure has been replaced with a lighter version which just
focusses on PAHs as suggested. This is much clearer and highlights the regions of the
chromatogram dominated by species in each phase.

[Figure]

Figure 1. Gas and particle phase composition of PAH emissions from burning cow dung cake.

Figure 7 and line 548-550 suggest that more of the mass could be speciated with new
instruments. However Figure 7 focuses on peaks that can be positively matched with
something in one of the standards. Is it possible to infer composition based on the mass
spectra of the unidentified peaks?

This is tackled in the companion paper acp 2020-892, which may not have been visible at the
point of submission of this manuscript. High levels of total speciation are achieved (>90 %)
using other instruments. We will direct readers to acp 2020-892 in the text.

Quantitative emission factors of VOCs from the combustion of solid fuels characteristic to
Delhi are provided in a companion publication (Stewart et al., 2020b).

Grammatical comments: Line 152 and 153 - it seems like "samples" in the former line mean
the sampled media (filters and SPE), and in the latter it means the fuels. Please clarify.

This has been clarified throughout, with samples referring to the organic components
collected and analysed and fuel types referring to the biofuels collected and burnt.

Line 195 - define EtOAc Line 347 uses NVOC to indicate "non-volatile", but the paragraph
starting at line 389 seems to use NVOC to indicate "nitrogen-containing." Please clarify.

EtoAC and MeOH are now defined in the text as requested.

The reviewer makes a valid point about the use of NVOC as "non-volatile" and therefore the
nitrogen containing VOCs are no longer abbreviated as NVOCs.

Commented [JH1]:

Reviewer comment 2

Stewart et al. have developed a gas-chromatography-based analytical technique to speciate
and quantify semi-volatile and intermediate volatility organic compounds (S/IVOCs) and
applied this technique to measure emissions of S/IVOCs from domestic fuels used in Delhi,
India. S/IVOCs are important precursors to ozone and aerosol formation in the atmosphere
and there is need to develop robust analytical techniques to speciate and quantify their
emissions. Biomass burning is an important source of global air pollution and the type of
biomass burning studied here (i.e., biofuel combustion) is a particularly understudied
emission source. Hence, the work described in the manuscript is well motivated. I should also
commend the authors for a well written manuscript that provides all the necessary details to comment on the methods and the interpretation of the results. The analytical method development was well designed and the application was very well described, although I should note that I am not trained as an analytical chemist. This should serve as a useful resource for researchers doing similar work in the atmospheric community. The primary results of speciation and quantification are well described too but, given the large dataset that is being analyzed, only a small fraction of the data are actually presented. I recommend the publication of this study in Atmospheric Chemistry and Physics after the authors have had a chance to respond to my, mostly big-picture and minor, comments.

Big-picture comments: 1. The introduction seems too generic and long at the moment and needs to be realigned to describe the state-of-the-science and gaps as it relates to the key findings from this work. For instance, lines 53-68 discuss S/IVOC emissions generally but don't focus on those emissions from biomass burning.

The introduction has been rewritten, and shortened, as suggested with the new structure: (1) general introduction to organic emissions from biomass burning, (2) state-of-the-art studies focussed on SOA formation from biomass burning, (3) state-of-the-art studies focussed on GCxGC analysis of biomass burning emissions and the research gap to be filled, (4) state-of-the-art studies focussed on detailed I/SVOC analyses of south Asian fuels, (5) research gaps to be filled with this work in terms of analytical procedures and poorly understood, but widely used, Asian fuels.

General discussion of historical biomass burning I/SVOC measurement procedures, application of GCxGC to other source sectors and Indian air quality has now been removed. It now reads:

[revised manuscript text omitted]

Text from lines 69 to 107 could be condensed into a few sentences. Earlier work relevant to this paper seems to be mentioned in lines 108 to 145 and needs to be highlighted, front and center.

This has been rectified through the new introduction paragraph.

Another point that could be highlighted is that S/IVOC emissions are poorly, if at all, represented in emissions inventories and chemical transport models and their impacts on atmospheric chemistry and air quality are uncertain (with particular relevance to regions where this and similar fuel use is dominant, e.g., Asia).

This is now highlighted in the first paragraph of the introduction as:

I/SVOC emissions are poorly, if at all, represented in regional inventories and chemical transport models.

2. Given the large variability seen in biomass burning emissions, say relative to internal combustion engines, the authors should comment on the single experiments done for most of the fuels. This could be done by analyzing the experiment-to-experiment variability for the fuels where multiple experiments were done (i.e., cow dung cake, waste), as well as through a review of similar literature.

This is a great comment. We now acknowledge this in the text.

Figure 8 shows that there was large sample to sample variability in emission factors for different fuel wood samples, for which only 1 sample was taken. For this reason, emission factors have been generalised for use in budget estimates to the type of fuel. Mean emission factors are provided for measurements from samples of 17 fuel woods, 3 crop residues, 3 cow dung cakes and 3 different collections of municipal solid waste. Despite this, for LPG and charcoal samples only 1 sample was measured, and this significantly increases the uncertainty in the PAH emission factors from these fuel sources.

We also look at this in some more detail in the companion paper, acp 2020-892, but in a more quantitative manner, presenting total non-methane volatile organic compound emission factors and looking at over 50 different combustion experiments of fuelwoods This is better presented in this companion paper as there is a larger dataset of emissions.

In addition, they should also comment on the differences in combustion encountered in their setup versus a real-world application. For example, most municipal solid waste is probably burned in a high-temperature incinerator where the combustion chemistry might be very different than the combustion simulated in this work. I do understand that 'backyard' low-temperature MSW fires are a major concern in India, including in Delhi.

This is now acknowledged in the text as shown below.

This study was conducted under controlled laboratory conditions. For some sample types, such as municipal solid waste, the laboratory measurement may not be entirely reflective of real-world conditions. Municipal solid waste combustion may occur under both flaming and smouldering conditions at landfill sites and in backyards, as well as in high-temperature incinerators in more developed countries. All of these are likely to have quite different combustion chemistry, and consequently lead to varying levels of emission.

3. I commend the authors on putting together this fantastic dataset of speciation and emission factors and I am fairly certain that this will serve as a comprehensive resource for years to come (from studying exposure to toxic pollutants to developing accurate emissions inventories for air quality modeling). [This must be an oversight but I did not see a 'data availability' section that describes how and where the data will be archived for others to use]. However, the manuscript seems to present only a 'snapshot' of the dataset, with a mix of higher-level observations and depth for only a subset of speciated organic compounds (e.g., PAHs). Correct me if I am wrong but there is so much more to the dataset than what is presented. If that is indeed the case, what I would have liked to see is a structured vision for how the data plans to be analyzed further (e.g., detailed source profiles, molecular markers for source identification, volatility distributions) and what open, pressing questions would this dataset help answer in the long run?

We have added a table of peaks measured on both SPE disks and PTFE filters to the SI. We also now have the companion paper acp 2020-892, which gives speciated emission factors from these burns and provides quantitative emission factors in the supplementary information. We will provide a short reference within this paper to acp 2020-892 to emphasise this.

We have also submitted a paper to another journal which provides comprehensive organic emission profiles from 5 different instruments. This analyses the volatility distribution of organic emissions from $C_2$-$C_{40}$. The paper also provides additional insight into the new datasets by looking at the secondary organic aerosol production potential, OH reactivity and toxicity of emissions from domestic fuels in Delhi, India.

In addition, the dataset (provided in acp 2020-892) will also be used to better constrain NMVOC emissions through formation of local and regional emission inventories.

Minor comments: 1. Line 228: Explain what 'NIST library hit was >800' means.

This has now been explained in the text.

Peaks were assigned through comparison of retention times with known standards and
comparison with the National Institute of Standards and Technology (NIST) mass spectral
library. Peaks with no genuine standard available were tentatively identified if the NIST library
similarity was > 700. This provides an indication of how similar the mass spectra obtained was
to the database mass spectra for the peak, with more details given in Stein, (2011). Peaks with
a hit > 900 reflect an excellent match, 800-900 a good match and 700-800 a fair match (Stein,
2011). The uncertainty in this approach has been shown to be low for peaks of hits > 800, with
the probability of incorrect identification being around 30 % for hits between 800-900 and 14
% for matches above 900 (Worton et al., 2017).

2. Line 471-485: Clarifying questions. Is the low fraction of the speciation of the organic
aerosol limited to not finding a match in the NIST or does it highlight a problem with the
analytical method? In addition, how sensitive is the fraction speciated to the use of the filter
media, i.e., better with PTFE versus quartz?

The low fraction of speciation of organic aerosol is due to lack of genuine standards available
to develop detector-response curves for the mass spectrometer. Many peaks have a mass
spectral database hit, but this does not impact the percentage identified here.

Percentage identification should not be influenced by the filter sampling media. This is low in
complex samples due to lack of genuine standards to allow quantitative detector-response
curves for individual analytes to be developed. The accelerated solvent extraction method
used here may miss some of the most polar water soluble species as they are not soluble in
EtOAc.

3. The composition section could benefit from findings from some recent publications that
have studied SOA from biomass burning emissions or precursors, e.g., He et al. (ESPI, 2020) –
alkylfuran mixture, Joo et al., (ESC, 2019) – 3-methylfuran, Ahern et al. (JGR, 2019), Akherati
et al. (ES&T, 2020), and Lim et al. (ACP, 2019) – biomass burning SOA in laboratory
experiments with an emphasis on understanding phenolic, furanic, and monoterpene VOC
contributions to SOA.

This paper now includes references to the above papers which focus on SOA formation from biomass burning emissions. Some have been included in the adjusted introduction and some in the composition section.

Introduction:

Ahern et al. (2019) showed that for burning of biomass needles, biogenic VOCs were the dominant class of SOA precursor.

SOA formation from biomass burning has been shown to be significant in laboratory studies, with SOA yields from the burning of western U.S. fuels reported to be 24±4 % after 6 h and 56±9 % after 4 d (Lim et al., 2019).

Main body:

SOA formation from furanic species remains poorly understood, with a recent study showing an SOA yield of 1.6-2.4 % during the oxidation of 3-methylfuran with the nitrate radical (Joo et al., 2019).

A recent study found that, oxygenated aromatic compounds, which included phenols and methoxyphenols, were responsible for just under 60 % of the SOA formed from western U.S. fuels (Akherati et al., 2020).

4. Figure 7: Was total organic mass in the gas- and particle phase measured another way, e.g., FID-gas, Sunset OC/EC-particle, to get mass closure?

Organic mass in the gas phase was measured with two separate gas chromatographs and a proton-transfer reaction mass spectrometer, with the data treated in the companion paper acp 2020-892. No direct-FID measurements of total hydrocarbon were attempted, in part due to the difficulty of dealing with the change in sensitivity of the FID for hydrocarbons and OVOCs which constitute a large fraction of the gas phase VOCs. Organic mass balance is treated in a subsequent publication which maps emissions onto a volatility-basis dataset and has been submitted for review elsewhere. However, OC/EC was not measured on these PM samples due to the small sample size.

5. Figure 8: Mention sample size for each fuel. Specify measurement uncertainty when n=1. Combine measurement uncertainty and experiment-to-experiment variability when n>1.

This point has been addressed in major point 2.

Table 1 shows the sample size for each fuel. Figure 8 shows that there was large sample to sample variability in emission factors for different wood samples, for which only 1 sample was taken. For this reason, emission factors have been generalised for use in budget estimates to the type of fuel. Mean emission factors have been provided for measurements from samples of 17 fuel woods, 3 crop residues, 3 cow dung cakes and 3 different collections of municipal solid waste. The mean values and standard deviations of measured emission factors are as follows: fuelwood ($247 \pm 214$ mg kg$^{-1}$), crop residue ($747 \pm 518$ mg kg$^{-1}$), MSW ($1022 \pm 340$ mg kg$^{-1}$) and cow dung cake ($615 \pm 112$ mg kg$^{-1}$). Despite this, for LPG and charcoal samples only

1 sample was measured, and this significantly increases the uncertainty in the PAH emission factors from these fuel sources. Experiment to experiment variability is provided in the companion paper acp 2020-892.

6. Figure 9: Was the gas/particle partitioning of PAHs analyzed further? Seems like an ideal dataset to study absorptive partitioning.

This has not been analysed as part of this work. A subsequent publication will provide a volatility-basis dataset of emissions from the fuel types studied here. It should be possible to dilute this to real-world dilutions, particulate matter concentrations and temperatures to better understand, and represent, the gas/particle phase partitioning of emissions from biomass burning.

7. Lines 631- 632: Why were certain samples fully speciated and others not very much? Was this relationship examined further with respect to its sensitivity to fuel, total organic mass captured on filter/disc, other variables?

Certain fuels were not fully speciated due to the low quantity of organic matter released and the availability of standards to quantify these. For extremely complex samples, genuine standards were not available to develop quantitative mass-spectrometer response curves to these peaks and resulted in low levels of total quantification. Future suggestions are given in the text of splitting the eluent between the mass spectrometer and a flame ionisation detector to provide much better semi quantification.

**Marked up manuscript**

[revised manuscript text omitted]

A wide array of NVOCsnitrogen containing VOCs were present in the cow dung cake samples, with over 600 nitrogen containing peaks including aromatics such as peaks on SPE disks and PTFE filters (SPE/PTFE) from pyridines and pyrizines (123pyrazines (43/35), amines (8247/28), amides (7738/37), nitriles (74), 742/31), 6-membered heterocycles (1), 613/14), 5-membered heterocycles (28), 5 membered heterocycles including aromatics such as pyrolespyrroles as well as pyrolinespyrazolines and pyrolidines (97pyrrolidines (50/45), 4-membered heterocycles (6), 3/3), 3-membered heterocycles (64/1), nitrogen containing PAHs (38)14/24), imidazoles (229/12), imines (4), isocyanates (3), hydrazines (7), carbamic acids (3),/1) and azoles (33) oximines (3) and sulfur containing nitrogen compounds (1423/10).

Previous studies have measured the nitrogen content of cow dung cake to be as high as 1.9 % (Stockwell et al., 2014) in comparison to other fuel types such as fuel woods (0.14-0.35 %), rice straws (0.4 %) and coal (0.6 %). The large amount of NVOCsnitrogen containing VOCs are likely formed from the volatilisation and decomposition of nitrogen-containing compounds within the cow dung cake, such as free amino acids, pyrroline, pyridine and chlorophyll (Leppalahti and Koljonen, 1995; Burling et al., 2010; Ren and Zhao, 2015). NVOCsNitrogen containing VOCs are of concern because they can be extremely toxic (Ramírez et al., 2014; Farren et al., 2015) and amines in particular can change the hydrological cycle by leading to the creation of new particles (Smith et al., 2008; Kirkby et al., 2011; Yu and Luo, 2014) which act as cloud condensation nuclei (Kerminen et al., 2005; Laaksonen et al., 2005; Sotiropoulou et al., 2006).

Figure 6 shows a comparison of organic aerosol composition observed from different fuel types (LPG, fuel wood, sawdust and municipal solid waste). The measured emissions had very different compositions, reflecting the variability of organic components produced from different sample types.fuel types (see the Supplementary Information S7 and S8 for species observed from different sample types). Quantitative emission factors of VOCs from the combustion of solid fuels characteristic to Delhi are provided in a companion publication (Stewart et al., 2020b). 
[revised manuscript text omitted]